# CSG: Cognitive Structure Generation for Intelligent Education

**Hengnian Gu** [1 2] **Zhifu Chen** [1] **Yuxin Chen** [1] **Jin Peng Zhou** [3] **Dongdai Zhou** [1]

## Abstract

Cognitive structure (CS), a student's construction of concepts and inter-concept relations, has long been recognized as a foundational notion in psychology and intelligent education, yet remains largely unassessable in practice. Existing approaches such as knowledge tracing (KT) and cognitive diagnosis (CD) simplify and indirectly approximate CS, but they intertwine representation learning with prediction objectives, limiting generalization, interpretability, and reuse across tasks. To address this gap, we propose Cognitive Structure Generation (CSG), a task-agnostic framework that explicitly models CS through generative modeling. Based on educational theories, CSG first pretrains a Cognitive Structure Diffusion Probabilistic Model (CSDPM) and then applies reinforcement learning with SOLO-based hierarchical rewards to capture plausible patterns of cognitive development. By decoupling cognitive structure representation from downstream prediction, CSG produces interpretable and transferable cognitive structures that can be seamlessly integrated into diverse student modeling tasks. Experiments on five real-world datasets show that CSG yields more comprehensive representations, substantially improving performance while offering enhanced interpretability and modularity.

## 1. Introduction

Cognitive structure (CS), originally conceived in topological psychology and later embraced by cognitive psychology in education (Piaget, 1952; Bruner, 2009; Ausubel, 1968), denotes the knowledge system within a student's mind, man-

ifested as an inherent learning state. Through learning processes, students continually integrate new concepts and reorganize existing ones to refine their cognitive structures for further learning. Formally, a cognitive structure can be modeled as an evolving *graph* (Novak & Gowin, 1984), with nodes and edges representing the student's construction of concepts and relations, respectively (Steffe & Gale, 1995).

Cognitive structure assessment has long been a central topic in psychometrics (Lord & Novick, 2008) and has been widely applied in intelligent educational systems (Anderson et al., 2014). Traditional methods primarily relied on expert-defined educational principles to directly calculate cognitive structure but lacked sufficient accuracy (Tatsuoka, 2009; Lin et al., 2016b). Considering that cognitive structure is an inherent learning state, researchers have shifted to indirectly measuring it based on students' responses to test items. Knowledge tracing (KT) (Corbett & Anderson, 1994) and cognitive diagnosis (CD) (Leighton & Gierl, 2007) are prototypical tasks. KT predicts the response $r_{\tau+1}$ at time $\tau$ as $P_{KT}(r_{\tau+1}) = f_{KT}(\boldsymbol{h}_\tau, \boldsymbol{\beta}_{\tau+1}; \Phi)$, where $\boldsymbol{h}_\tau$ is the student's latent state inferred from historical interactions, $\boldsymbol{\beta}_{\tau+1}$ is the tested item's features, and $\Phi$ denotes the model parameters (Abdelrahman et al., 2023). CD models the association between response $r$ and student's cognitive state or ability $\boldsymbol{\theta}$ based on tested item $\beta$ as $P_{CD}(r) = f_{CD}(\boldsymbol{\theta}, \boldsymbol{\beta}; \Omega)$, where $\Omega$ denotes the model parameters (Wang et al., 2024). Although recently emerged KT (Piech et al., 2015; Choi et al., 2020; Zhang et al., 2017) and CD (Cheng et al., 2019; Wang et al., 2020) models have achieved remarkable performance, they still face two foundational limitations.

First, both the student's latent state $\boldsymbol{h}_\tau$ in KT and the cognitive state or ability $\boldsymbol{\theta}$ in CD are typically narrowed to the student's construction of individual concepts, i.e. $\boldsymbol{h}_\tau, \boldsymbol{\theta} \rightarrow \mathbb{R}^L$ (where $L$ is the number of concepts), and thus cannot model the student's construction of inter-concept relations necessary for modeling a complete cognitive structure and its holistic evolution during the real learning process. Although some studies have applied graph learning methods on static concept maps (Liu et al., 2019; Nakagawa et al., 2019; Tong et al., 2020) or heterogeneous interaction graphs (Gao et al., 2021; Yang et al., 2024) to obtain enhanced representations of $\boldsymbol{h}_\tau$ and $\boldsymbol{\theta}$, they only model students' construction on individual concepts and still do not explicitly model students' construction of inter-concept relations. Moreover,

[1]School of Information Science and Technology, Northeast Normal University, Changchun, Jilin, China [2]Undergraduate School of Artificial Intelligence, Shenzhen Polytechnic University, Shenzhen, Guangdong, China [3]Cornell University, Ithaca, New York, United States. Correspondence to: Dongdai Zhou <ddzhou@nenu.edu.cn>.

*Proceedings of the 43rd International Conference on Machine Learning*, Seoul, South Korea. PMLR 306, 2026. Copyright 2026 by the author(s).

many graph-augmented approaches implicitly rely on an expert-specified knowledge structure (e.g., concept graphs / Q-matrices), whose misspecification can directly propagate into biased learner-specific cognitive structures.

Thus, our core motivation is to explicitly and comprehensively model cognitive structure, the states of the students' construction of concepts and inter-concept relations (Ausubel, 1968), without relying on expert-specified knowledge structures that risk introducing misspecification bias. Despite its foundational role in learning theory, such a construct remains largely unassessable in educational practice.

Second, by definition, students' responses are only an external manifestation or an indirect indicator of their underlying learning state—namely, the cognitive structure in this paper, $h_\tau$ in KT, and $\theta$ in CD. Yet most existing models have become increasingly preoccupied with maximizing response prediction accuracy, often through extensive domain feature integration (Liu et al., 2021; Xu et al., 2023; Zhou et al., 2021), ever more sophisticated network designs and optimizations (Yang et al., 2023a;b; Li et al., 2024; Liu et al., 2024b; Chen et al., 2023), and so forth. While such directions improve accuracy, they still tightly couple state inference with prediction, intertwining representation learning with predicted objectives, which restricts generalization, particularly when models are applied in cold-start or uncertain settings, and limits interpretability and modular reuse.

To bridge this gap, we propose **Cognitive Structure Generation (CSG)**, a task-agnostic framework that explicitly models CS through generative modeling, which decouples cognitive structure representation from downstream prediction. Guided by cognitive structure theory (Ausubel, 1968) and constructivism (Steffe & Gale, 1995), CSG aims to produce interpretable and transferable cognitive structures that can be seamlessly integrated into diverse student modeling tasks, thereby enhancing generalization, interpretability, and modularity. Specifically:

**First**, considering that a cognitive structure is manifested as a graph, we naturally cast *cognitive structure generation* as a *graph generation* task, and propose a *Cognitive Structure Diffusion Probabilistic Model* (CSDPM), whose forward diffusion and reverse denoising processes can learn the underlying distribution of real cognitive structures and produce novel ones. However, since real cognitive structures cannot be directly observed, we devise a rule-based method to infer students' construction of concepts and inter-concept relations from interaction logs, yielding a set of simulated cognitive structures, which is then used to pretrain the CSDPM and initialize its basic capability for CSG.

**Second**, although the cognitive structures sampled from the pretrained CSDPM match the distribution over simulated cognitive structures, they are insufficient to reflect the gen-

uine levels of cognitive development (Flavell, 1977; Keil, 1992) that students achieve through their learning processes. To fill this gap, inspired by *the Structure of the Observed Learning Outcome (SOLO) taxonomy* (Biggs et al., 2014) that characterizes five levels of cognitive development, we define a fine-grained, hierarchical reward function. Using these reward signals, we optimize the policy of the denoising process via reinforcement learning to better capture plausible patterns of cognitive development.

To this end, the pretrained and fine-tuned CSDPM has been fully equipped for cognitive structure generation, and the generated cognitive structures can be leveraged for diverse downstream student modeling tasks in the educational domain. To the best of our knowledge, we are the **first** to: **(i)** reformulate cognitive structure modeling as a cognitive structure generation task; **(ii)** decouple cognitive structure representation from downstream prediction; **(iii)** propose a CSDPM with a two-stage design, pretraining on simulated structures and fine-tuning via reinforcement learning with SOLO-based hierarchical rewards. Experimental results on five datasets show that cognitive structures generated by CSG offer more comprehensive and effective representations for student modeling, substantially improving performance on KT and CD tasks while enhancing interpretability.

**Conflict of Interest Disclosure.** The authors declare no financial conflicts of interest related to this work.

## 2. Related Works

We organize related works into three strands. **Cognitive Structure Modeling** has been rooted in psychology and education (Piaget, 1952; Ausubel, 1968), where traditional psychometric approaches construct rule-based graphs of students' concepts and relations but lack personalization. With the rise of learning analytics, researchers approximate cognitive structures from student responses via knowledge tracing (Piech et al., 2015; Choi et al., 2020) and cognitive diagnosis (Leighton & Gierl, 2007; Cheng et al., 2019). KT methods employ hidden-state models, classifiers, or encoder–decoders, sometimes augmented with concept maps or heterogeneous graphs (Liu et al., 2019; Yang et al., 2024), while CD methods focus on fine-grained attributes (Xu et al., 2023). We also note recent diffusion-based KT/CD models such as MSKT (Zhang et al., 2024b) and DiffCog (Zhao et al., 2024), which couple diffusion processes with latent knowledge representations for improved KT/CD prediction. However, they mainly focus on students' mastery of individual concepts while overlooking their construction of inter-concept relations, thereby limiting the modeling of holistic cognitive-structure evolution. Recent attempts still rely on predefined graphs (Chen et al., 2024; Gu et al., 2025a) and are often tailored to a single task, instead of learning a unified latent cognitive structure representation

that can be reused across downstream tasks (e.g., knowledge tracing and cognitive diagnosis), leaving holistic cognitive structure generation largely unexplored. These limitations motivate a generative perspective, namely directly generating learner-specific cognitive structures, especially inter-concept relation graphs, from student interaction data. **Graph Diffusion Probabilistic Models (DPMs)** extend deep generative frameworks such as autoregressive models, VAEs, GANs, and normalizing flows. Continuous-time DPMs (Jo et al., 2022) denoise Gaussian-corrupted graphs, whereas discrete variants (Vignac et al., 2023) use categorical transitions to better preserve sparsity. Moreover, compared with VAE-style graph generators, diffusion provides a more flexible framework for cognitive-structure synthesis. These advances demonstrate the potential of diffusion models for complex graph generation (Vignac et al., 2023; Kong et al., 2023). However, existing diffusion-based KT/CD methods (Zhang et al., 2024b; Zhao et al., 2024) mainly diffuse latent mastery states and do not explicitly generate personalized inter-concept relation graphs. Meanwhile, generic graph DPMs are typically not conditioned on or aligned with educational interaction semantics, leaving their mechanisms to be adapted for the unique challenges of cognitive structure generation. **Optimization of DPMs** has increasingly leveraged reinforcement learning to align generative models with external objectives. Recent approaches in vision (Fan et al., 2023; Black et al., 2024) and graphs (Liu et al., 2024c) treat reverse diffusion as a Markov decision process optimized via policy gradients. Building on this line of work, we propose a SOLO-based reward to optimize the graph diffusion model for CSG, thereby aligning the generated structures more effectively with cognitive development levels. For a more comprehensive discussion of related studies, please refer to Appendix A.

## 3. The CSG Framework

### 3.1. Problem Formulation

Suppose a learning system is defined as $\mathcal{L} = \langle S, Q, K, R \rangle$, where $S = \{s_i\}_{i=1}^{N}$ is the set of $N$ students, $Q = \{q_j\}_{j=1}^{M}$ the set of $M$ questions, and $K = \{k_l\}_{l=1}^{L}$ the set of $L$ knowledge concepts. Students answer questions from $Q$, producing response logs. For each student $s_i$, let $j_{i,\tau} \in \{1, \ldots, M\}$ denote the index of the question attempted at interaction $\tau$, and let $r_{i,\tau} \in \{0, 1\}$ be the observed response (1 for correct and 0 otherwise). The interaction history of student $s_i$ up to interaction $n$ is denoted as $X_i^{(n)} = \{(q_{j_{i,\tau}}, r_{i,\tau})\}_{\tau=1}^{n}$, where $\tau$ indexes interaction time.

A student $s_i$'s cognitive structure at interaction $n$ is defined as a graph $\mathcal{G}_i^{(n)} = (\mathcal{V}_i^{(n)}, \mathcal{E}_i^{(n)})$. The node set $\mathcal{V}_i^{(n)} \in \mathbb{R}^{L \times c}$ represents $s_i$'s construction states for the $L$ concepts in $K$, and the edge set $\mathcal{E}_i^{(n)} \in \mathbb{R}^{L \times L \times c}$ represents the construc-

tion states of inter-concept relations, where $c$ is the size of the discrete construction state space (e.g., "constructed" vs. "unconstructed"). Since we treat the cognitive structure as an undirected graph, all subsequent operations are applied to the upper-triangular entries $\mathcal{E}^+$ of $\mathcal{E}$, after which the matrix is symmetrized. Our goal is to generate $\mathcal{G}_i^{(n)}$ from $X_i^{(n)}$, formally defined as a mapping function $f_{CSG} : X_i^{(n)} \to \mathcal{G}_i^{(n)}$.

To implement this mapping, we propose the *Cognitive Structure Diffusion Probabilistic Model* (CSDPM). The CS-DPM is first pretrained on simulated cognitive structures to initialize its generative capacity, and then fine-tuned via policy optimization to align generation with genuine cognitive development. The holistic structures produced by the optimized CSDPM can then be used in downstream tasks such as knowledge tracing (KT) and cognitive diagnosis (CD): $P_{KT}(r_{i,n+1}) = f_{KT}(\mathcal{G}_i^{(n)}, \boldsymbol{\beta}(q_{j_{i,n+1}}); \Phi)$ and $P_{CD}(r_{ij}) = f_{CD}(\mathcal{G}_i^{(n)}, \boldsymbol{\beta}(q_j); \Omega)$, where $\boldsymbol{\beta}(q)$ denotes the embedding of question $q$, and $\Phi, \Omega$ are model parameters.

The overall architecture of CSG is illustrated in Fig.1, which consists of two stages: pretraining CSDPM and optimizing CSDPM, which we will detail in the following subsections.

### 3.2. Stage I: Pretraining CSDPM with Simulated Cognitive Structures

The goal of Stage I is to initialize the CSDPM so that it captures meaningful inductive biases about how students construct knowledge. Unlike other graph generation domains (Liu et al., 2024a; Zhang et al., 2024a; Trivedi et al., 2024), training here ideally requires access to ground-truth cognitive structures, which are not directly observable in practice. To address this, we design a simple *rule-based simulation process* grounded in theories of cognitive structure (Ausubel, 1968) and constructivist learning (Steffe & Gale, 1995), which serves as a proxy for pretraining.

**Cognitive Structure Simulation.** For each student $s_i$ and interaction history $X_i^{(n)}$, we simulate a cognitive structure $\tilde{\mathcal{G}}_i^{(n)} = (\mathcal{V}_i^{(n)}, \mathcal{E}_i^{(n)})$ by defining rule-based functions for concept states and relation states. Inspired by (Lin et al., 2016a), we compute the construction state of concept $k_l$ by

$$f_{UOC}(k_l, X_i^{(n)}) = \frac{\sum_{\tau=1}^{n} \omega_{l, j_{i,\tau}} \cdot r_{i,\tau}}{\sum_{\tau=1}^{n} \omega_{l, j_{i,\tau}}}, \tag{1}$$

and the construction state of the relation between concepts $k_a$ and $k_b$ by

$$f_{UOR}(k_a, k_b, X_i^{(n)}) = \frac{\sum_{\tau=1}^{n} \mathbf{1}\{\omega_{a, j_{i,\tau}} > 0 \wedge \omega_{b, j_{i,\tau}} > 0\}(\omega_{a, j_{i,\tau}} + \omega_{b, j_{i,\tau}}) r_{i,\tau}}{\sum_{\tau=1}^{n} \mathbf{1}\{\omega_{a, j_{i,\tau}} > 0 \wedge \omega_{b, j_{i,\tau}} > 0\}(\omega_{a, j_{i,\tau}} + \omega_{b, j_{i,\tau}})}, \tag{2}$$

where $\omega_{l,j}$ denotes the weight of concept $k_l$ in question $q_j$, obtained by normalizing the Q-matrix across concepts that a question involves. This ensures that if a question

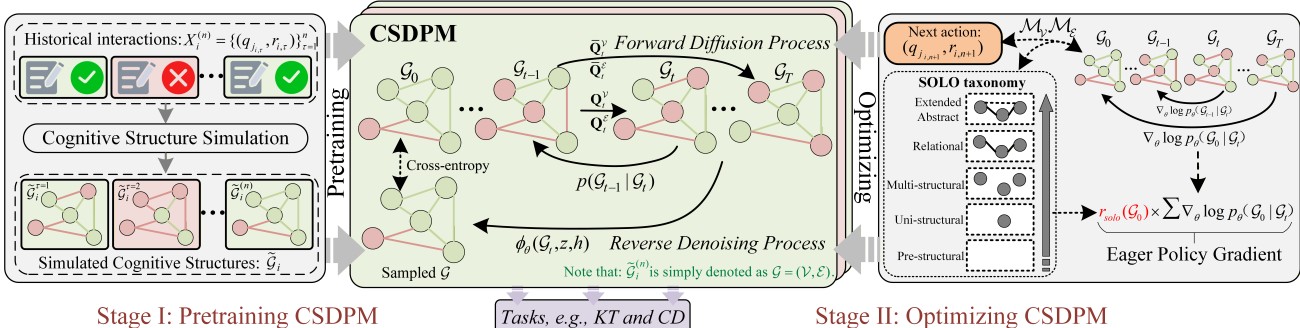

*Figure 1.* **(Overview).** The CSG includes two stages: pretraining CSDPM with Simulated Cognitive Structures and optimizing CSDPM via SOLO-based Hierarchical Reward. In stage I, the Cognitive Structure Simulation module (left) produces simulated cognitive structures that are used to pretrain the CSDPM. In stage II, a SOLO-based reward is introduced to optimize the CSDPM's policy via RL (right). Once pretrained and optimized, the CSDPM can generate cognitive structures, whose effectiveness is validated on KT and CD tasks through response prediction.

taps multiple concepts, each receives a proportional share of weight. To better reflect real-world data and improve robustness, we also add small Gaussian perturbations to the Q-matrix entries. In Appendix H, we also provide a detailed example with full calculation steps.

**Intuition.** Equations 1 and 2 can be viewed as *weighted accuracies* that approximate the likelihood a student has constructed a given concept or relation. Eq. 1 averages the student's correctness on all questions involving concept $k_l$, weighted by how strongly the question tests $k_l$. Intuitively, if a student answers many $k_l$-related questions correctly, the ratio will approach 1, signaling that the concept is well constructed. Eq. 2 measures co-construction: it averages correctness on questions that involve *both $k_a$ and $k_b$*, weighted by their combined relevance. Thus, if a student tends to succeed on joint questions, the relation between the two concepts is considered constructed.

**From probabilities to discrete states.** The $f_{UOC}$ and $f_{UOR}$ are empirical probabilities in $[0, 1]$. To map them into the discrete construction space $\Delta^c$, we round the values and apply a one-hot encoding, yielding $\tilde{v}_{i,l}^{(n)}$ and $\tilde{e}_{i,a-b}^{(n)}$. By repeating this process for all students $s_i$ and interaction indices $n$, we obtain a set of simulated cognitive structures $\tilde{\mathbb{G}}$, which provides the training data to pretrain the CSDPM through forward diffusion and reverse denoising. For clarity, we drop student subscripts and interaction superscripts when unambiguous, writing $\mathcal{G}, v, e$ in place of $\tilde{\mathcal{G}}_i^{(n)}, \tilde{v}_{i,l}^{(n)}, \tilde{e}_{i,a-b}^{(n)}$.

**Forward Diffusion Process**. Our CSDPM uses a forward diffusion process $q(\mathcal{G}_{1:T} \mid \mathcal{G}_0) = \prod_{t=1}^{T} q(\mathcal{G}_t \mid \mathcal{G}_{t-1})$ that gradually corrupts an initial simulated cognitive structure $\mathcal{G}_0 \sim q(\mathcal{G}_0)$ into near–uniform noise $q(\mathcal{G}_T)$ after $T$ steps. The transition admits a node/edge factorization over the discrete construction state space:

$$q(\mathcal{V}_t \mid \mathcal{V}_{t-1}) = \prod_{v \in \mathcal{V}} q(v_t \mid v_{t-1}),$$
$$q(\mathcal{E}_t \mid \mathcal{E}_{t-1}) = \prod_{e \in \mathcal{E}^+} q(e_t \mid e_{t-1}). \tag{3}$$

where $\mathcal{E}^+$ denotes the upper–triangular edge set (the graph is symmetrized afterwards). For each categorical node state $v \in \Delta^c$, we use the discrete noising kernel $q(v_t|v_{t-1}) = Cat(v_t; v_{t-1}\mathbf{Q}_t^v), \mathbf{Q}_t^v = \alpha_t \mathbf{I} + (1-\alpha_t)\frac{\mathbf{1}_c\mathbf{1}_c^\top}{c}$ with schedule $\alpha_t \in [0,1]$ decreasing as $t$ increases (Austin et al., 2021). Here, $\mathbf{1}_c$ is the $c$-dimensional all-ones vector and $\frac{\mathbf{1}_c\mathbf{1}_c^\top}{c}$ is the uniform transition over $\Delta^c$. Thus, $\alpha_t = 1$ leaves the signal unchanged ($\mathbf{Q}_t^v = \mathbf{I}$), while smaller $\alpha_t$ mixes in more uniform noise. Let $\bar{\mathbf{Q}}_t^v = \mathbf{Q}_1^v \mathbf{Q}_2^v \cdots \mathbf{Q}_t^v$. Then the marginal and one–step posteriors admit closed forms:

$$q(v_t \mid v_0) = \mathrm{Cat}(v_t; v_0 \bar{\mathbf{Q}}_t^v),$$
$$q(v_{t-1} \mid v_t, v_0) = \mathrm{Cat}\left(v_{t-1}; \frac{(v_t(\mathbf{Q}_t^v)^\top) \odot (v_0 \bar{\mathbf{Q}}_{t-1}^v)}{v_0 \bar{\mathbf{Q}}_t^v v_t^\top}\right), \tag{4}$$

where $\odot$ denotes element-wise product and all vectors are row-stochastic. As $t$ grows and $\prod_{s=1}^{t} \alpha_s \to 0$, each node approaches the uniform distribution $q(v_T \mid v_0) \approx \mathrm{Cat}(v_T; \frac{\mathbf{1}_c}{c})$; edge transitions are defined analogously.

**Reverse Denoising Process**. Given the forward corruption, we learn a parametric reverse process $p_\theta(\mathcal{G}_{0:T}) = p(\mathcal{G}_T) \prod_{t=1}^{T} p_\theta(\mathcal{G}_{t-1} \mid \mathcal{G}_t)$ to recover cognitive structures from near–uniform noise $p(\mathcal{G}_T) \approx q(\mathcal{G}_T)$. We factor the reverse transition into nodes and edges:

$$p_\theta(\mathcal{G}_{t-1}|\mathcal{G}_t) = \prod_{v \in \mathcal{V}} p_\theta(v_{t-1}|\mathcal{G}_t) \prod_{e \in \mathcal{E}^+} p_\theta(e_{t-1}|\mathcal{G}_t). \tag{5}$$

Following the standard $x_0$-parameterization in discrete diffusion (Hasselt, 2010; Karras et al., 2022), each conditional can be expressed by marginalizing the exact posterior with a prediction of the clean state:

$$p_\theta(v_{t-1}|\mathcal{G}_t) = \sum_{v_0 \in \Delta^c} q(v_{t-1}|v_t, v_0)\, p_\theta(v_0|\mathcal{G}_t),$$
$$p_\theta(e_{t-1}|\mathcal{G}_t) = \sum_{e_0 \in \Delta^c} q(e_{t-1}|e_t, e_0)\, p_\theta(e_0|\mathcal{G}_t), \tag{6}$$

where a neural network predicts $p_\theta(v_0 | \mathcal{G}_t)$ and $p_\theta(e_0 | \mathcal{G}_t)$ given the noisy graph $\mathcal{G}_t$.

**Training Objective**. We pretrain on the simulated dataset $\tilde{\mathbb{G}}$ by maximizing the expected log-likelihood of clean structures conditioned on noisy ones:

$$J_{\text{CSDPM}}(\theta) = \mathbb{E}_{\mathcal{G}_0 \sim \tilde{\mathbb{G}}, t \sim \mathcal{U}[\![1,T]\!]} \left[ \mathbb{E}_{q(\mathcal{G}_t|\mathcal{G}_0)} \left[ \log p_\theta(\mathcal{G}_0|\mathcal{G}_t) \right] \right], \quad (7)$$

with $t$ sampled uniformly from $[\![1,T]\!]$. At generation time, we sample $\mathcal{G}_T \sim p(\mathcal{G}_T)$ and iteratively draw $\mathcal{G}_{t-1} \sim p_\theta(\mathcal{G}_{t-1}|\mathcal{G}_t)$ to obtain the trajectory $(\mathcal{G}_T, \mathcal{G}_{T-1}, \ldots, \mathcal{G}_0)$.

**Parametrization**. We instantiate $p_\theta$ with an extended Graph Transformer (Dwivedi & Bresson, 2021; Vignac et al., 2023) that takes a noisy cognitive structure $\mathcal{G}_t = (\mathcal{V}_t, \mathcal{E}_t)$ as input and outputs distributions over clean node and edge states. Following (Vignac et al., 2023), we retain graph-theoretic feature integration and condition the model on two auxiliary features: (i) a diffusion-step embedding that encodes the current noise level $t$, and (ii) an embedding of the student's interaction history $X^{(n)}$, which provides task-specific guidance. An algorithmic summary is provided in Appendix B.

While the simulation provides a theory-grounded proxy for pretraining, it may introduce heuristic bias; we calibrate the generator via reward–driven policy optimization in Stage II.

### 3.3. Stage II: Optimizing CSDPM via SOLO-based Hierarchical Reward

Building on the pretrained CSDPM, we further optimize its reverse denoising process to better align generation with genuine cognitive development. Inspired by the SOLO taxonomy (Biggs et al., 2014), we introduce a fine-grained hierarchical reward function and cast the denoising process as a reinforcement learning problem.

**Standard Markov Decision Process Formulation**. A standard MDP is specified by $(\mathcal{S}, \mathcal{A}, \mathcal{P}, r, \rho_0)$, where $\mathcal{S}$ is the state space, $\mathcal{A}$ the action space, $\mathcal{P}(s'|s, a)$ the transition kernel, $r(s, a)$ the reward, and $\rho_0$ the initial-state distribution. Under a parameterized policy $\pi_\theta(a|s)$, an agent generates a trajectory $\boldsymbol{\xi} = (s_0, a_0, \ldots, s_T)$ by sampling $s_0 \sim \rho_0$, then repeatedly choosing $a_t \sim \pi_\theta(\cdot|s_t)$, receiving reward $r(s_t, a_t)$, and transitioning via $s_{t+1} \sim \mathcal{P}(\cdot|s_t, a_t)$. The return is $\mathcal{R}(\boldsymbol{\xi}) = \sum_{t=0}^{T} r(s_t, a_t)$, and the RL objective is to maximize $\mathcal{J}_{\text{RL}}(\theta) = \mathbb{E}_{\boldsymbol{\xi} \sim p(\boldsymbol{\xi}|\pi_\theta)}[\mathcal{R}(\boldsymbol{\xi})]$. By the policy-gradient theorem (Grondman et al., 2012), this objective can be optimized using REINFORCE(Sutton et al., 1998):

$$\nabla_\theta \mathcal{J}_{\text{RL}}(\theta) = \mathbb{E}_{\boldsymbol{\xi} \sim p(\boldsymbol{\xi}|\pi_\theta)} \left[ \sum_{t=0}^{T} \nabla_\theta \log \pi_\theta(a_t|s_t) \mathcal{R}(\boldsymbol{\xi}) \right]. \quad (8)$$

**Mapping the Reverse Denoising Process to a $T$-step MDP**. The pretrained CSDPM defines samples via its reverse denoising chain $p_\theta(\mathcal{G}_{0:T})$, but the marginal $p_\theta(\mathcal{G}_0)$ is intractable (Ho et al., 2020), and the reward $r(\mathcal{G}_0)$ is a black box with no gradient signal (Black et al., 2024). Following (Fan et al., 2023; Liu et al., 2024c), we reformulate the

denoising process as a $T$-step MDP:

$$\begin{aligned}
&\boldsymbol{s}_t \triangleq (\mathcal{G}_{T-t}, T - t), \quad \boldsymbol{a}_t \triangleq \mathcal{G}_{T-t-1}, \\
&\pi_\theta(\boldsymbol{a}_t|\boldsymbol{s}_t) \triangleq p_\theta(\mathcal{G}_{T-t-1}|\mathcal{G}_{T-t}, T - t), \\
&\mathcal{P}(\boldsymbol{s}_{t+1}|\boldsymbol{s}_t, \boldsymbol{a}_t) \triangleq \delta(\boldsymbol{s}_{t+1} - (\mathcal{G}_{T-t-1}, T - t - 1)), \\
&r(\boldsymbol{s}_t, \boldsymbol{a}_t) \triangleq r(\mathcal{G}_0) \text{ if } t = T, \quad r(\boldsymbol{s}_t, \boldsymbol{a}_t) \triangleq 0 \text{ if } t < T,
\end{aligned} \quad (9)$$

where $\delta(\cdot)$ denotes a Dirac distribution, capturing the fact that transitions are deterministic: given $s_t$ and $a_t$, the next state is exactly $s_{t+1} = (\mathcal{G}_{T-t-1}, T - t - 1)$. The initial state $s_0 = (\mathcal{G}_T, T)$ is the fully noised structure, and the terminal state $s_T = (\mathcal{G}_0, 0)$ is the fully denoised structure.

**SOLO-based Hierarchical Reward Function**. After formulating the reverse denoising process of CSDPM as a MDP, we can optimize it for specific reward signals, which should ideally reflect the levels of cognitive development that students achieve through their learning processes. Inspired by the SOLO taxonomy (Biggs et al., 2014), we propose a fine-grained, hierarchical reward function that scores the generated cognitive structures according to their alignment with the five levels of SOLO, which correspond to progressively better construction of concepts and inter-concept relations within more sophisticated cognitive structure.

Given a sampled structure $\mathcal{G}_0 = (\mathcal{V}_0, \mathcal{E}_0)$ and the next real interaction $(q_{j_{i,n+1}}, r_{i,n+1})$, we compare the predicted construction of relevant concepts and relations against the observed response. The matching degrees are:

$$\mathcal{M}_\mathcal{V} = \frac{1}{|\mathcal{V}_{q_{j_{i,n+1}}}|} \sum_{v \in \mathcal{V}_{q_{j_{i,n+1}}}} (r_{i,n+1} \veebar v), \mathcal{M}_\mathcal{E} = \frac{1}{|\mathcal{E}_{q_{j_{i,n+1}}}|} \sum_{e \in \mathcal{E}_{q_{j_{i,n+1}}}} (r_{i,n+1} \veebar e),$$

where $\veebar$ denotes the XNOR operation. The SOLO-based reward is then

$$r_{solo}(\mathcal{G}_0) = \begin{cases}
r_1, & \mathcal{M}_\mathcal{V} = 0, \\
r_2, & 0 < \mathcal{M}_\mathcal{V} < \kappa, \\
r_3, & \mathcal{M}_\mathcal{V} \geq \kappa \wedge \mathcal{M}_\mathcal{E} < \kappa, \\
r_4, & \kappa \leq \mathcal{M}_\mathcal{V} < 1 \wedge \kappa \leq \mathcal{M}_\mathcal{E} < 1, \\
r_5, & (\mathcal{M}_\mathcal{V} = 1 \wedge \mathcal{M}_\mathcal{E} \geq \kappa) \\
& \vee(\mathcal{M}_\mathcal{V} \geq \kappa \wedge \mathcal{M}_\mathcal{E} = 1),
\end{cases} \quad (10)$$

with $r_1 < r_2 < r_3 < r_4 < r_5$ corresponding to SOLO levels: (i) *Pre-structural*: No meaningful concept alignment; (ii) *Uni-structural*: Alignment of a single or few concepts; (iii) *Multi-structural:* Alignment of multiple concepts, few relations; (iv) *Relational*: Alignment of multiple concepts and multiple relations; (v) *Extended abstract*: Alignment of almost all concepts and relations.

Since $\mathcal{M}_\mathcal{V}, \mathcal{M}_\mathcal{E} \in [0, 1]$, we adopt $\kappa = 0.5$ as the default threshold to distinguish "few" from "multiple" alignments. For instance, $0 < \mathcal{M}_\mathcal{V} < 0.5$ maps to the uni-structural level and is rewarded with $r_2$. Sensitivity analyses on thresholds and reward scales are reported in Appendix F.

**Policy Gradient Estimation.** With the denoising process formulated as a $T$-step MDP, an agent generates a CSG

trajectory $\boldsymbol{\xi} = (\mathcal{G}_T, \mathcal{G}_{T-1}, \ldots, \mathcal{G}_0)$, where $\boldsymbol{\xi} \sim p(\boldsymbol{\xi}|\pi_\theta) = p_\theta(\mathcal{G}_{0:T})$. Since rewards are only assigned at the terminal state, the cumulative return of any trajectory reduces to

$$\mathcal{R}(\boldsymbol{\xi}) = \sum_{t=0}^{T} r(\boldsymbol{s}_t, \boldsymbol{a}_t) = r_{solo}(\mathcal{G}_0). \quad (11)$$

The learning objective is therefore $\mathcal{J}_{\mathrm{RL}}(\theta) = \mathbb{E}_{\boldsymbol{\xi} \sim p(\boldsymbol{\xi}|\pi_\theta)}[\mathcal{R}(\boldsymbol{\xi})] = \mathbb{E}_{\mathcal{G}_{0:T} \sim p_\theta}[r_{solo}(\mathcal{G}_0)]$, which coincides with the end-structure objective $\mathcal{J}_{\mathcal{G}_0}(\theta)$.

A standard REINFORCE estimator gives the gradient

$$\nabla_\theta \mathcal{J}_{\mathrm{RL}}(\theta) = \mathbb{E}_{\mathcal{G}_{0:T} \sim p_\theta} \left[ r_{solo}(\mathcal{G}_0) \sum_{t=1}^{T} \nabla_\theta \log p_\theta(\mathcal{G}_{t-1}|\mathcal{G}_t) \right], \quad (12)$$

but this estimator suffers from high variance on discrete graph diffusion. Following Liu et al. (2024c), we instead adopt the *eager policy gradient*, which directly reinforces the likelihood of high-reward terminal structures (i.e., the clean cognitive structures after $T$ reverse denoising steps), rather than distributing credit iteratively via the term $\nabla_\theta \log p_\theta(\mathcal{G}_{t-1}|\mathcal{G}_t)$. With Monte Carlo estimation, the policy gradient can be modified as follows:

$$\nabla_\theta \mathcal{J}_{\mathrm{RL}}(\theta) \approx \frac{1}{|\mathcal{D}|} \sum_{d=1}^{|\mathcal{D}|} \frac{T}{|\mathcal{T}_d|} \sum_{t \in \mathcal{T}_d} r_{solo}(\mathcal{G}_0^{(d)}) \nabla_\theta \log p_\theta(\mathcal{G}_0^{(d)} | \mathcal{G}_t^{(d)}). \quad (13)$$

where $\mathcal{D}$ is the set of sampled trajectories, and $\mathcal{T}_d \subseteq [\![1, T]\!]$ is a random subset of timesteps for trajectory $d$. This estimator treats all trajectories ending at the same $\mathcal{G}_0$ as an equivalence class and reinforces them jointly, which significantly improves stability and sample efficiency. The full policy optimization procedure is summarized in Appendix C.

**Two-Stage Training Paradigm.** Overall, the training of CSG adopts a two-stage paradigm, inspired by the pretraining–finetuning strategy of LLMs (Devlin et al., 2019). In Stage I, it bypasses pure noise by leveraging simulated cognitive structures grounded in educational principles to establish a meaningful prior. In Stage II, a SOLO-based hierarchical reward assesses the generated structures by how well they match the progressive levels of understanding defined by the SOLO, which guides CSG to refine its initial representations and move beyond handcrafted assumptions.

## 4. Experiments

**Downstream Modeling for CSG.** Since ground-truth cognitive structures cannot be directly observed, we follow the standard evaluation approach in prior work (Piech et al., 2015; Wang et al., 2020) and use learning performance outcomes as an indicator of latent representation quality. The basic idea is that if the generated structures capture students' latent cognitive states, the resulting representations should improve prediction accuracy on standard benchmarks. We focus on two widely studied tasks: *knowledge tracing* (KT), which predicts learning performance, and *cognitive diagnosis* (CD), which estimates fine-grained knowledge proficiency. Together, these tasks serve as proxies for assessing

how well the structures encode interpretable and transferable cognitive information.

**From Structures to Representations.** To operationalize the generated cognitive structures in downstream models, we employ the *edge-aware hard-clustering graph pooling* method from Zhu et al. (2023). This method produces a compact cognitive state vector for each student by jointly summarizing node and edge features, thereby preserving information about both concept mastery and inter-concept relation mastery. The resulting vector is concatenated with the tested question embedding before being passed to the task-specific output layers.

**CSG-KT.** For knowledge tracing, we use the pooled structure representation to augment a standard DKT (Piech et al., 2015) model. The prediction function is

$$P_{KT}(r_{i,n+1}) = \sigma\Big(\mathrm{FC}\Big(\mathrm{Pooling}(\mathcal{G}_i^{(n)}) \oplus \mathrm{emb}(\boldsymbol{\beta}(q_{j_{i,n+1}}))\Big)\Big). \quad (14)$$

where $n$ is the current interaction index, $\mathrm{emb}(\cdot)$ denotes the question embedding, $\oplus$ is concatenation, FC is a fully-connected layer, and $\sigma$ is the sigmoid activation. This formulation allows the model to predict whether student $s_i$ will answer the next question $q_{j_{i,n+1}}$ correctly, informed by their generated cognitive structure.

**CSG-CD.** For cognitive diagnosis, we integrate the pooled structure representation into the NCD framework (Wang et al., 2020). The prediction function is

$$P_{CD}(r_{ij}) = \sigma\Big(\boldsymbol{\mathcal{Q}}_j \odot \Big((\mathrm{Pooling}(\mathcal{G}_i^{(n)}) - \boldsymbol{h}_{diff}) \times \boldsymbol{h}_{disc}\Big)\Big), \quad (15)$$

where $\boldsymbol{\mathcal{Q}}_j$ is one row of the Q-matrix that specifies which concepts question $q_j$ assesses. The vectors $\boldsymbol{h}_{diff}$ and $\boldsymbol{h}_{disc}$ are transformations of the question embedding $\mathrm{emb}(\boldsymbol{\beta}(q_j))$, following Wang et al. (2020). Here, $\odot$ and $\times$ denote element-wise product and scalar multiplication, respectively. This formulation assesses the consistency of a student's structure $\mathcal{G}_i^{(n)}$ with an observed response $r_{ij}$. Both CSG-KT and CSG-CD are trained via cross-entropy loss.

**Experimental Settings.** We evaluate our CSG on five real-world datasets of varying scales: Math1, Math2, FrcSub, NIPS34, and the larger ASSISTment17[1], with statistics provided in Appendix D. To evaluate the utility of the cognitive structures, we compare CSG-KT and CSG-CD against several baselines in their respective tasks. For KT, we include DKT (Piech et al., 2015), SAKT (Pandey & Karypis, 2019), GKT (Nakagawa et al., 2019), SKT (Tong et al., 2020), GRKT (Cui et al., 2024), MIKT (Sun et al., 2024), ENAS-

---

[1]Math1, Math2, FrcSub are available at http://staff.us tc.edu.cn/~qiliuql/data/math2015.rar. NIPS34 is available at http://ednet-leaderboard.s3-websi te-ap-northeast-1.amazonaws.com/.ASSISTment17 is available at https://sites.google.com/view/assi stmentsdatamining/dataset?authuser=0

*Table 1.* Performance comparison between CSG-KT and CSG-CD with their baselines on different datasets, averaged over five-fold cross-validation. Statistical significance is assessed via the Wilcoxon rank-sum test, with * ($p < 0.05$), ** ($p < 0.01$), and *** ($p < 0.001$).

| Task | Model | Math1 | | | Math2 | | | FrcSub | | | NIPS | | | ASSISTment17 | | |
|---|---|---|---|---|---|---|---|---|---|---|---|---|---|---|---|---|
| | Metrics | AUC↑ | ACC↑ | RMSE↓ | AUC↑ | ACC↑ | RMSE↓ | AUC↑ | ACC↑ | RMSE↓ | AUC↑ | ACC↑ | RMSE↓ | AUC↑ | ACC↑ | RMSE↓ |
| KT | DKT | 0.7735 | 0.7082 | 0.4524 | 0.7381 | 0.6678 | 0.4600 | 0.8202 | 0.7529 | 0.3392 | 0.6593 | 0.6214 | 0.4690 | 0.6832 | 0.6683 | 0.4543 |
| | SAKT | 0.7612 | 0.7017 | 0.4552 | 0.7250 | 0.6583 | 0.4618 | 0.8113 | 0.7513 | 0.3419 | 0.6531 | 0.6176 | 0.4710 | 0.6762 | 0.6703 | 0.4601 |
| | GKT | 0.7843 | 0.7147 | 0.4493 | 0.7463 | 0.6759 | 0.4519 | 0.8247 | 0.7608 | 0.3360 | 0.6841 | 0.6339 | 0.4645 | 0.7226 | 0.7092 | 0.4502 |
| | SKT | 0.7895 | 0.7181 | 0.4489 | 0.7529 | 0.6842 | 0.4492 | 0.8385 | 0.7696 | 0.3338 | 0.6985 | 0.6429 | 0.4637 | 0.7349 | 0.7242 | 0.4457 |
| | GRKT | 0.7943 | 0.7242 | 0.4461 | 0.7618 | 0.6976 | 0.4448 | 0.8418 | 0.7754 | 0.3280 | 0.7070 | 0.6501 | 0.4601 | 0.7493 | 0.7359 | 0.4426 |
| | MIKT | 0.8030 | 0.7281 | 0.4412 | 0.7701 | 0.7017 | 0.4426 | 0.8472 | 0.7804 | 0.3253 | 0.7147 | 0.6570 | 0.4583 | 0.7677 | 0.7481 | 0.4403 |
| | ENAS-KT | 0.8103 | 0.7326 | 0.4334 | 0.7722 | 0.7120 | 0.4405 | 0.8506 | 0.7865 | 0.3207 | 0.7233 | 0.6634 | 0.4565 | 0.7812 | 0.7627 | 0.4384 |
| | simpleKT | 0.8074 | 0.7304 | 0.4390 | 0.7713 | 0.7083 | 0.4411 | 0.8485 | 0.7844 | 0.3232 | 0.7191 | 0.6618 | 0.4542 | 0.7791 | 0.7605 | 0.4389 |
| | PSI-KT | 0.8118 | 0.7392 | 0.4317 | 0.7759 | 0.7140 | 0.4403 | 0.8533 | 0.7908 | 0.3309 | 0.7260 | 0.6687 | 0.4520 | 0.7890 | 0.7653 | 0.4337 |
| | DiffuQKT | 0.8116 | 0.7347 | 0.4333 | 0.7716 | 0.7098 | 0.4406 | 0.8519 | 0.7889 | 0.3225 | 0.7300 | 0.6695 | 0.4510 | 0.7756 | 0.7579 | 0.4394 |
| | **CSG-KT** | **0.8220**\** | **0.7412**\** | **0.4283**\** | **0.7772**\* | **0.7197**\* | **0.4390**\* | **0.8636**\** | **0.8022**\** | **0.3192**\** | **0.7413**\** | **0.6757**\** | **0.4511**\** | **0.7963**\** | **0.7792**\** | **0.4313**\** |
| CD | IRT | 0.7356 | 0.7179 | 0.4279 | 0.7589 | 0.6981 | 0.4516 | 0.7414 | 0.7091 | 0.3944 | 0.7489 | 0.6907 | 0.4516 | 0.7381 | 0.6847 | 0.4694 |
| | MIRT | 0.7482 | 0.7347 | 0.4256 | 0.7699 | 0.7038 | 0.4478 | 0.8086 | 0.7745 | 0.3589 | 0.7589 | 0.7017 | 0.4483 | 0.7558 | 0.6950 | 0.4546 |
| | NCD | 0.7691 | 0.7459 | 0.4084 | 0.7781 | 0.7182 | 0.4456 | 0.8250 | 0.8042 | 0.3498 | 0.7697 | 0.7113 | 0.4412 | 0.7687 | 0.7064 | 0.4462 |
| | RCD | 0.7861 | 0.7584 | 0.4033 | 0.7911 | 0.7275 | 0.4406 | 0.8321 | 0.8178 | 0.3419 | 0.7736 | 0.7171 | 0.4345 | 0.7769 | 0.7157 | 0.4431 |
| | HyperCDM | 0.7876 | 0.7599 | 0.4016 | 0.7972 | 0.7320 | 0.4383 | 0.8417 | 0.8239 | 0.3387 | 0.7821 | 0.7209 | 0.4301 | 0.7927 | 0.7301 | 0.4387 |
| | DisenGCD | 0.7983 | 0.7628 | 0.4001 | 0.8039 | 0.7457 | 0.4324 | 0.8559 | 0.8375 | 0.3342 | 0.7886 | 0.7311 | 0.4275 | 0.7842 | 0.7249 | 0.4392 |
| | DiffCog | 0.7892 | 0.7115 | 0.4403 | 0.7766 | 0.7038 | 0.4421 | 0.8623 | 0.7873 | 0.3928 | 0.7755 | 0.7098 | 0.4401 | 0.7772 | 0.7083 | 0.4419 |
| | **CSG-CD** | **0.8133**\* | **0.7710** | **0.3987**\*** | **0.8179**\** | **0.7521**\* | **0.4270**\*** | **0.8699**\*** | **0.8451**\** | **0.3152**\*** | **0.8036**\*** | **0.7507**\** | **0.4242**\*** | **0.8003**\*** | **0.7386**\** | **0.4320**\** |

*Table 2.* Ablation study on the impact of CSG variants for KT and CD across multiple datasets.

| Task | Model | Math1 | | | Math2 | | | FrcSub | | | NIPS | | | ASSISTment17 | | |
|---|---|---|---|---|---|---|---|---|---|---|---|---|---|---|---|---|
| | Metrics | AUC↑ | ACC↑ | RMSE↓ | AUC↑ | ACC↑ | RMSE↓ | AUC↑ | ACC↑ | RMSE↓ | AUC↑ | ACC↑ | RMSE↓ | AUC↑ | ACC↑ | RMSE↓ |
| KT | $V_1$-KT | 0.7842 | 0.7050 | 0.4496 | 0.7276 | 0.6745 | 0.4571 | 0.8144 | 0.7486 | 0.3455 | 0.6807 | 0.6504 | 0.4697 | 0.7388 | 0.7521 | 0.4589 |
| | $V_2$-KT | 0.7991 | 0.7196 | 0.4433 | 0.7421 | 0.6887 | 0.4543 | 0.8288 | 0.7630 | 0.3397 | 0.6951 | 0.6647 | 0.4674 | 0.7611 | 0.7605 | 0.4515 |
| | $V_3$-KT | 0.8042 | 0.7343 | 0.4472 | 0.7567 | 0.6930 | 0.4511 | 0.8433 | 0.7775 | 0.3241 | 0.7196 | 0.6691 | 0.4663 | 0.7789 | 0.7679 | 0.4497 |
| | $V_4$-KT | 0.8085 | 0.7351 | 0.4413 | 0.7614 | 0.6974 | 0.4491 | 0.8479 | 0.7821 | 0.3287 | 0.7242 | 0.6697 | 0.4604 | 0.7810 | 0.7702 | 0.4410 |
| | $V_5$-KT | 0.8111 | 0.7387 | 0.4322 | 0.7758 | 0.7184 | 0.4379 | 0.8598 | 0.7882 | 0.3262 | 0.7318 | 0.6730 | 0.4528 | 0.7855 | 0.7749 | 0.4329 |
| | **CSG-KT** | **0.8220** | **0.7412** | **0.4283** | **0.7772** | **0.7197** | **0.4390** | **0.8636** | **0.8022** | **0.3192** | **0.7413** | **0.6757** | **0.4511** | **0.7963** | **0.7792** | **0.4313** |
| CD | $V_1$-CD | 0.7870 | 0.7477 | 0.4218 | 0.7967 | 0.7277 | 0.4508 | 0.8210 | 0.8063 | 0.3475 | 0.7671 | 0.7068 | 0.4411 | 0.7669 | 0.7022 | 0.4545 |
| | $V_2$-CD | 0.7913 | 0.7520 | 0.4157 | 0.8008 | 0.7319 | 0.4471 | 0.8354 | 0.8138 | 0.3309 | 0.7713 | 0.7210 | 0.4371 | 0.7707 | 0.7194 | 0.4489 |
| | $V_3$-CD | 0.7958 | 0.7665 | 0.4098 | 0.8051 | 0.7463 | 0.4406 | 0.8601 | 0.8385 | 0.3276 | 0.7857 | 0.7254 | 0.4313 | 0.7851 | 0.7218 | 0.4449 |
| | $V_4$-CD | 0.7965 | 0.7669 | 0.4041 | 0.8086 | 0.7469 | 0.4395 | 0.8650 | 0.8434 | 0.3275 | 0.7903 | 0.7300 | 0.4257 | 0.7892 | 0.7270 | 0.4429 |
| | $V_5$-CD | 0.7985 | 0.7673 | 0.4030 | 0.8169 | 0.7473 | 0.4377 | 0.8661 | 0.8438 | 0.3205 | 0.7997 | 0.7392 | 0.4353 | 0.7968 | 0.7311 | 0.4431 |
| | **CSG-CD** | **0.8133** | **0.7710** | **0.3987** | **0.8179** | **0.7521** | **0.4270** | **0.8699** | **0.8451** | **0.3152** | **0.8036** | **0.7507** | **0.4242** | **0.8003** | **0.7386** | **0.4320** |

KT (Yang et al., 2023a), simpleKT (Liu et al., 2023), PSI-KT (Zhou et al., 2024b), and DiffuQKT (Yu et al., 2025). For CD, we include IRT (Cai et al., 2016), MIRT (Ackerman et al., 2003), NCD (Wang et al., 2020), RCD (Gao et al., 2021), HyperCDM (Shen et al., 2024), DisenGCD (Yang et al., 2024), and DiffCog (Zhao et al., 2024).

Our goal is to broadly evaluate the utility of CSG-generated structures as general-purpose representations across tasks, rather than to provide an exhaustive benchmark of all KT/CD models. We therefore select representative baselines from three main categories: (i) classical models (e.g., DKT, IRT), (ii) structure-aware models (e.g., GKT, SKT, GRKT, RCD), and (iii) recent state-of-the-art models (e.g., simpleKT, PSI-KT, ENAS-KT, HyperCDM, DisenGCD, DiffuQKT, DiffCog). Following common practice, we use AUC (Bradley, 1997), ACC, and RMSE as evaluation metrics. Additional implementation details can be found in Appendix E. The code is available at https://github.com/shida-edu4ai/CSDPM.

All experiments use an 8:1:1 random split of interaction records. This split is strictly *disjoint*: no test interaction ever appears in the training set, and no model is trained on test data. For evaluation, CSG generates cognitive structures from a student's interaction history up to time $n$. For KT,

these structures are used to predict the response at $n+1$. For CD, the model is never exposed to the target response $r_{ij}$ for the item it is asked to predict. This ensures that evaluation strictly measures generalization rather than memorization, and that no information leaks from training to testing.

**Overall Performance.** Table 1 reports the performance of CSG-KT, CSG-CD, and all KT/CD baselines on five public datasets, measured by average AUC, ACC, and RMSE over 5-fold cross-validation, with the best scores highlighted in bold. We observe: **(i)** CSG-KT not only substantially outperforms classical knowledge tracing models (e.g., DKT, SAKT), but also delivers clear gains over graph-based methods that model only concept construction without inter-concept relations (e.g., GKT, SKT, GRKT), and even surpasses recent SOTA approaches (e.g., PSI-KT, MIKT, ENAS-KT). Similarly, CSG-CD markedly improves upon classical parameter-estimation models (e.g., IRT, MIRT), achieves significant gains over neural models that represent student ability only at the concept level (e.g., NCD), and also exceeds heterogeneous graph-based SOTA methods. These results indicate that generative cognitive structures provide more comprehensive and accurate representations of student learning states, while capturing their dynamic evolution over time. **(ii)** Across datasets of very different

*Table 3.* Detailed configurations of CSG variants.

| Variants | Pretraining | Optimization | |
|---|---|---|---|
| | | $r(\cdot)$ | $r_{solo}(\cdot)$ |
| $V_1$ | ✗ | ✗ | ✗ |
| $V_2$ | ✓ | ✗ | ✗ |
| $V_3$ | ✗ | ✓ | ✗ |
| $V_4$ | ✗ | ✗ | ✓ |
| $V_5$ | ✓ | ✓ | ✗ |
| CSG | ✓ | ✗ | ✓ |

scales and interaction densities, both CSG-KT and CSG-CD consistently deliver robust performance, underscoring the general applicability of our framework. We note that we employed simple KT/CD models with CSG to demonstrate effectiveness and reduce confounding factors, leaving adaptation to advanced methods for future work.

**Ablation Study.** We evaluate several variants of our framework by comparing their prediction performance on sampled cognitive structures, as summarized in Table 3: **(i)** $V_1$ uses only the rule-based simulated structures without any learning; **(ii)** $V_2$ pretrains CSDPM on simulated structures but does not apply RL optimization; **(iii)** $V_3$ skips pretraining and applies RL with a generic reward $r(\cdot)$; **(iv)** $V_4$ skips pretraining and applies RL with the SOLO-based reward $r_{solo}(\cdot)$; **(v)** $V_5$ combines pretraining with RL under the generic reward; and **(vi)** CSG is our complete framework with both pretraining and SOLO-based optimization. The generic reward $r(\cdot)$ does not differentiate developmental levels and simply sums $\mathcal{M}_\mathcal{V}$ and $\mathcal{M}_\mathcal{E}$ into a single scalar.

For a fair comparison, we use the rule-based simulated set $\tilde{\mathbb{G}}$ for $V_1$, and sample the corresponding generated set $\mathbb{G}_0$ for variants $V_2$–$V_5$. Each variant is then used to independently train and evaluate downstream KT and CD models, denoted as $V_i$-**KT** and $V_i$-**CD**, respectively, for $i = 1, \ldots, 5$.

Results in Table 2 show several key findings: **(i)** Overall, performance steadily improves from the simplest variant $V_1$ through $V_5$ to our full CSG, for both KT and CD tasks. **(ii)** Despite its simplicity, $V_1$ performs competitively with classical baselines (e.g., DKT for KT, IRT and NCD for CD), validating that our rule-based simulation already provides a strong approximation of students' learning states. On Math1, Math2, and FrcSub, where sequences are short but coverage is high, this simulation is especially effective; on NIPS34, longer interaction sequences offset lower coverage, yielding similarly strong outcomes. **(iii)** $V_3$ generally outperforms $V_2$, suggesting that task-driven RL optimization can capture hidden learning patterns and incorporate them into generated structures. **(iv)** The improvements of $V_4$ over $V_3$, and of full CSG over $V_5$, highlight the value of explicitly modeling developmental levels and confirm the effectiveness of SOLO-based hierarchical rewards. Additional analyses on hyperparameters and inference time are provided in Appendix F, G.

**Visualization and Interpretability Analysis.** In this work,

interpretability is one of our main motivations for explicitly modeling cognitive structures. Specifically, past methods typically encode knowledge mastery or proficiency implicitly within model parameters and then rely on heatmaps or radar charts to visualize and interpret hidden states. Our CSG takes a step toward improving interpretability by constructing cognitive structures in line with cognitive structure theory (Ausubel, 1968) and constructivism (Steffe & Gale, 1995). In our CSG, nodes directly represent students' constructed states of knowledge concepts, and edges represent their constructed states of inter-concept relations, so that only minimal modification is needed for post-hoc analysis.

As shown in Fig. 2, we observe: **(i)** Subfigure (a) shows the cognitive structure generated by CSG-CD for student $s_5$ immediately before answering question $q_1$ (assessing concepts $k_{0,2,5,7,9}$). The student exhibits weak construction of both individual concepts and their inter-concept relations, so CSG-CD predicts that the student will answer incorrectly. (b) shows the structure for student $s_{18}$ before the same question $q_1$; here the student has strong construction of all five concepts but still weak construction of their relations, and CSG-CD again predicts that the student will answer incorrectly. (c) shows the structure for student $s_{37}$ before $q_1$; in this case, the student demonstrates strong construction of both concepts and relations, so CSG-CD predicts a correct response. **(ii)** (d) shows five representative cognitive structures generated by CSG-KT for student $s_{15}$ at different points in their learning trajectory. Over time, $s_{15}$'s cognitive structure evolves from minimal construction to a fully developed structure that integrates the entire knowledge system in $s_{15}$'s mind, broadly aligning with the SOLO taxonomy levels of cognitive development. These case studies illustrate that CSG-generated structures not only capture students' subjective construction of the objective knowledge system but also trace its evolution throughout learning. The results are consistent with established findings in educational psychology, thereby providing meaningful explanations for students' response behaviors.

**Human Study.** Beyond qualitative cases, since ground-truth cognitive structures are unavailable, we provide a quantitative evaluation of interpretability by comparing model-generated cognitive structures with human expert annotations. We asked annotators with MS and PhD backgrounds in education to label the cognitive-structure graphs of 25 randomly selected students across 20 interactions each (500 graphs) for both the FrcSub and Math2 datasets. These annotations serve as a proxy for ground-truth cognitive structures. For LLM-generated structures, we prompted GPT-5 and Llama-3-70B to produce binary adjacency matrices. We then evaluated both our CSG outputs and the LLM-generated graphs using Jaccard similarity and Graph Edit Distance (GED) against the human annotations. The results (Table 4) show that CSG consistently achieves higher

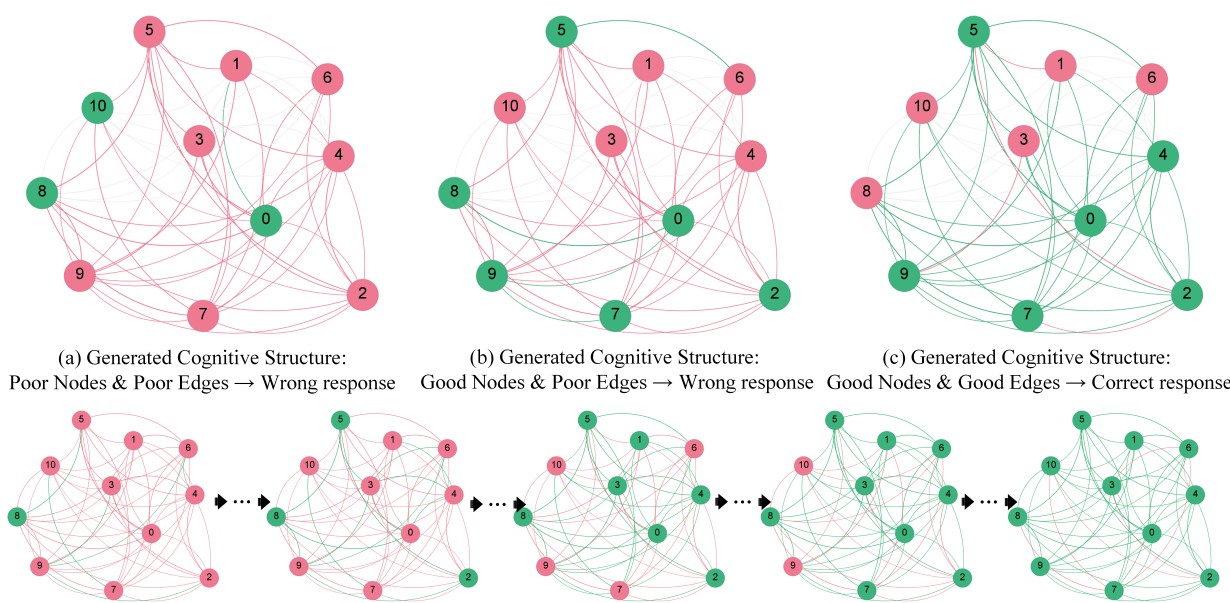

(a) Generated Cognitive Structure:
Poor Nodes & Poor Edges → Wrong response

(b) Generated Cognitive Structure:
Good Nodes & Poor Edges → Wrong response

(c) Generated Cognitive Structure:
Good Nodes & Good Edges → Correct response

(d) Evolution of a student's generated cognitive structure over five time points.

*Figure 2.* **(Case).** Examples of generated cognitive structures and the evolution process. Each graph depicts a student's generated cognitive structure at a given timestamp. Nodes represent the student's construction of concepts (the names of all concepts are listed in Table 11 in Appendix), while edges represent their construction of inter-concept relations. Green indicates fully constructed elements, red indicates elements not yet constructed, and gray denotes low-frequency or irrelevant edges shown for clarity.

*Table 4.* Structure-level interpretability evaluation on FrcSub and Math2. We report Jaccard similarity↑ and Graph Edit Distance (GED)↓ between generated graphs and human expert annotations.

| Proxy & Generator | FrcSub | | Math2 | |
|---|---|---|---|---|
| | Jaccard↑ | GED↓ | Jaccard↑ | GED↓ |
| Random | 0.18 | 0.87 | 0.14 | 0.89 |
| GPT-5 | 0.65 | 0.30 | 0.54 | 0.43 |
| Llama-3-70B | 0.39 | 0.61 | 0.34 | 0.67 |
| Simulated CS | 0.46 | 0.39 | 0.45 | 0.54 |
| **CSG** | **0.79** | **0.15** | **0.69** | **0.21** |

Jaccard similarity and lower GED, indicating that the cognitive structures it produces are more faithful to human-interpretable patterns. This provides quantitative evidence supporting the interpretability of CSG. The results also show that Stage-I simulated CS already has non-trivial agreement with human judgment, indicating an informative rather than arbitrary initialization; more importantly, Stage II substantially improves it and outperforms LLMs.

To further evaluate the quality of the learned cognitive structures, we compare CSG with LLM-generated structures on downstream KT and CD tasks. For each of the FrcSub and Math2 datasets, we randomly sample 500 students and prompt GPT-5 and Llama-3-70B to generate cognitive-structure graphs using the same annotation protocol. KT and CD models are then trained on the resulting graphs under identical optimization settings. As shown in Table 5, CSG consistently outperforms both LLM-based baselines across all KT and CD metrics, with particularly clear gains on KT. This suggests that CSG better captures the temporal

*Table 5.* Downstream KT/CD performance on 500-student subsets when using cognitive-structure graphs generated by different approaches. All models are trained with identical optimization procedures; only the input cognitive structures differ.

| Task | Method | FrcSub | | | Math2 | | |
|---|---|---|---|---|---|---|---|
| | | AUC↑ | ACC↑ | RMSE↓ | AUC↑ | ACC↑ | RMSE↓ |
| KT | GPT-5-KT | 0.7430 | 0.6683 | 0.3754 | 0.6785 | 0.6027 | 0.4840 |
| | Llama-3-70B-KT | 0.5170 | 0.4869 | 0.4139 | 0.4595 | 0.4030 | 0.5025 |
| | **CSG-KT** | **0.8602** | **0.8010** | **0.3197** | **0.7457** | **0.6854** | **0.4283** |
| CD | GPT-5-CD | 0.7858 | 0.7545 | 0.3462 | 0.7452 | 0.6759 | 0.4683 |
| | Llama-3-70B-CD | 0.5569 | 0.5376 | 0.3856 | 0.5063 | 0.4381 | 0.5077 |
| | **CSG-CD** | **0.8691** | **0.8433** | **0.3168** | **0.7885** | **0.7247** | **0.4389** |

evolution and task-relevant structure of student cognition. Although strong on general benchmarks, Llama-3-70B does not produce CS that yield competitive KT/CD performance.

## 5. Conclusion

We propose Cognitive Structure Generation (CSG), a graph diffusion framework for modeling students' evolving cognitive structures. CSG decouples structure representation from downstream prediction to generate explicit cognitive structures aligned with developmental patterns. It is trained via theory-grounded pretraining and SOLO-based reinforcement learning. Experiments on five datasets show consistent improvements in knowledge tracing and cognitive diagnosis, with enhanced generalizability, interpretability, and modularity. These results position holistic cognitive structure modeling as a foundation for transparent educational intelligence systems. Further discussion appears in Appendix J.

## Impact Statement

This paper presents work whose goal is to advance the field of machine learning in the education domain. There are many potential societal consequences of our work, none of which we feel must be specifically highlighted here.

## Acknowledgements

This work was supported by the National Natural Science Foundation of China (Grant No. 62477005), the Humanities and Social Science Fund of the Ministry of Education of China (Grant No. 24YJA880104), and the Jilin Provincial Scientific and Technological Development Key R&D Program (Grant No. 20230201063GX).

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

# A. Additional Discussion of Related Works

As a central topic in educational measurement, modeling cognitive structures has long remained a challenging task. With the advancement of educational data mining techniques, recent progress in graph generation offers promising support. Accordingly, we review related works as follows: *cognitive structure modeling*, *graph diffusion probabilistic models*, and *optimization of DPMs*.

**Cognitive Structure Modeling**. The students' cognitive structures (Lewin, 2013; Piaget, 1952; Bruner, 2009; Ausubel, 1968) represent their internal knowledge system, an evolving graph whose nodes reflect their construction of concepts and whose edges capture their construction of inter-concept relations (Novak & Gowin, 1984; Steffe & Gale, 1995). Traditional psychometric approaches derive such structures from expert-defined rules, which limit personalization and accuracy (Lord & Novick, 2008; Tatsuoka, 2009; Lin et al., 2016b). Considering that cognitive structure is an inherent learning state, researchers have shifted to indirectly measuring it based on students' responses to test items, e.g., knowledge tracing (KT) and cognitive diagnosis (CD).

From the KT perspective (Piech et al., 2015; Choi et al., 2020; Zhang et al., 2017), cognitive structures are implicitly approximated via students' learning states (also termed hidden states or knowledge states) inferred from response logs. This includes theory-guided state models (Gu et al., 2025b; Sun et al., 2024), mastery pattern classifiers (Briggs & Circi, 2017; Cui et al., 2016), and encoder–decoder architectures (Li et al., 2024; Liu et al., 2024b; Chen et al., 2023). Some KT methods enrich these states with static concept maps or heterogeneous interaction graphs (Liu et al., 2019; Nakagawa et al., 2019; Tong et al., 2020; Gao et al., 2021; Yang et al., 2024), yet they typically emphasize concept mastery without modeling the formation of inter-concept relations.

From the CD perspective (Leighton & Gierl, 2007; Cheng et al., 2019; Wang et al., 2020), models aim to identify fine-grained cognitive attributes or abilities underlying observed responses. While some approaches introduce additional features (Liu et al., 2021; Xu et al., 2023; Zhou et al., 2021), address data distribution issues (Cheng et al., 2025; Zhang et al., 2023b), or optimize network structures (Yang et al., 2023a;b), they also tend to focus on the correctness of individual concepts, overlooking the holistic evolution of cognitive structures.

Recent work has also coupled diffusion models with KT/CD objectives. DiffuQKT (Yu et al., 2025) and MSKT (Zhang et al., 2024b) uses a diffusion process to improve question representation and refine sequential latent knowledge states along student interaction logs for KT, and DiffCog (Zhao et al., 2024) applies diffusion as a denoiser over latent CD ability vectors to obtain more robust proficiency estimates; however, both operate purely in the latent-vector space and do not generate explicit, learner-specific cognitive structure graphs. Recent attempts still rely on predefined graphs (Chen et al., 2024; Gu et al., 2025a) and are often tailored to a single task, instead of learning a unified latent cognitive structure representation that can be reused across downstream tasks (e.g., knowledge tracing and cognitive diagnosis), leaving holistic cognitive structure generation largely unexplored. These limitations motivate a generative perspective, namely directly generating learner-specific cognitive structures, especially inter-concept relation graphs, from student interaction data. To our knowledge, we are the first to explicitly formulate the task of cognitive structure generation and present a unified framework for its holistic modeling.

**Graph Diffusion Probabilistic Models**. Graph generation has long relied on traditional deep generative frameworks (e.g., auto-regressive models (Liao et al., 2019), VAEs (Liu et al., 2018), GANs (Martinkus et al., 2022), and normalizing flows (Luo et al., 2021)) to capture complex graph distributions. More recently, diffusion probabilistic models (DPMs) (Ho et al., 2020) have emerged as a powerful new trend for graph generation (Zhang et al., 2023a). Continuous-time graph DPMs (e.g., EDP-GNN (Niu et al., 2020), GDSS (Jo et al., 2022), DruM (Jo et al., 2023)) learn to denoise Gaussian-corrupted graph representations (Song et al., 2021) but can struggle to preserve graph sparsity. To address this, discrete diffusion methods like DiGress (Vignac et al., 2023) replace continuous noise with categorical transitions, achieving strong results on complex benchmarks. Moreover, compared with VAE-style graph generators, diffusion provides a more flexible framework for cognitive-structure synthesis. These advances demonstrate the potential of diffusion models for complex graph generation (Vignac et al., 2023; Kong et al., 2023). To our knowledge, we are the first to introduce a graph diffusion probabilistic model for CSG.

**Optimization of DPMs**. Reinforcement learning (RL) has been widely used to steer graph generators toward downstream objectives. Traditional methods (Sutton et al., 1999; Zhou et al., 2018) rely on custom environments and exhibit high computational costs. Diffusion models (DPMs) have been aligned to external rewards in vision: DPO (Fan et al., 2023) and DDPO (Black et al., 2024) treat the reverse diffusion as a Markov decision process and apply policy gradients to optimize

black-box reward signals, and DPM alignment has been extended to graphs by GDPO (Liu et al., 2024c), which introduces an eager policy gradient. Thus, we propose a SOLO-based reward to optimize the CSDPM, which is effective for aligning with cognitive development levels.

## B. The complete procedure of pretraining CSDPM

---
**Algorithm 1** Pretraining CSDPM
---

1: **Input:** Simulated dataset $\tilde{\mathbb{G}}$, diffusion steps $T$, loss weight $\lambda_{ve}$
2: **while** not converged **do**
3:     Sample $(\mathcal{G}_0, X^{(n)}) \sim \tilde{\mathbb{G}}$                 ▷ Sample a simulated cognitive structure and its interaction sequence
4:     Sample $t \sim \mathcal{U}[\![1, T]\!]$
5:     Sample $\mathcal{G}_t \sim q(\mathcal{G}_t \mid \mathcal{G}_0)$
6:     $z \leftarrow f(\mathcal{G}_t, t)$             ▷ Graph-theoretic features
7:     $h \leftarrow \text{emb}(X^{(n)})$         ▷ Interaction-guidance features
8:     $(\hat{p}^{\mathcal{V}}, \hat{p}^{\mathcal{E}}) \leftarrow \phi_\theta(\mathcal{G}_t, z, h)$         ▷ Denoising pass
9:     optimizer.step$(\mathcal{L}_{CE}(\hat{p}^{\mathcal{V}}, \mathcal{V}_0) + \lambda_{ve}\mathcal{L}_{CE}(\hat{p}^{\mathcal{E}}, \mathcal{E}_0))$         ▷ Cross-entropy loss
10: **end while**

---

## C. The complete procedure of Policy Optimization

---
**Algorithm 2** Optimizing CSDPM
---

1: **Input:** Pretrained CSDPM $p_\theta$, diffusion steps $T$, reward function $r_{solo}(\cdot)$, learning rate $\eta$, number of trajectories $|\mathcal{D}|$, timestep samples $|\mathcal{T}|$, training steps $N$
2: **Output:** Optimized CSDPM $p_\theta$
3: **for** $n = 1, \ldots, N$ **do**
4:     **for** $d = 1, \ldots, |\mathcal{D}|$ **do**
5:         Sample cognitive structure trajectory $\mathcal{G}_{0:T}^{(d)} \sim p_\theta(\mathcal{G}_{0:T})$
6:         Compute reward $r_{solo}(\mathcal{G}_0^{(d)})$
7:         Sample random timesteps subset $\mathcal{T}_d \subseteq [\![1, T]\!]$
8:     **end for**
9:     $\bar{r} \leftarrow \frac{1}{|\mathcal{D}|}\sum_{d=1}^{|\mathcal{D}|} r_{solo}(\mathcal{G}_0^{(d)}), \quad \text{std}[r] \leftarrow \sqrt{\frac{1}{|\mathcal{D}|-1}\sum_{d=1}^{|\mathcal{D}|}(r_{solo}(\mathcal{G}_0^{(d)}) - \bar{r})^2}$         ▷ Estimate reward statistics
10:     $\nabla_\theta J_{\text{RL}}(\theta) \leftarrow \frac{1}{|\mathcal{D}|}\sum_{d=1}^{|\mathcal{D}|}\frac{T}{|\mathcal{T}_d|}\sum_{t \in \mathcal{T}_d}\frac{r_{solo}(\mathcal{G}_0^{(d)})-\bar{r}}{\text{std}[r]}\nabla_\theta \log p_\theta(\mathcal{G}_0^{(d)}|\mathcal{G}_t^{(d)})$         ▷ Estimate eager policy gradient
11:     $\theta \leftarrow \theta + \eta \cdot \nabla_\theta J_{\text{RL}}(\theta)$         ▷ Update parameters
12: **end for**

---

## D. Statistics of all five datasets.

*Table 6.* Statistics of all five datasets.

| Datasets | Math1 | Math2 | FrcSub | NIPS34 | ASSISTment17 |
|---|---|---|---|---|---|
| # of students | 4,209 | 3,911 | 536 | 4,918 | 1,678 |
| # of questions | 20 | 20 | 20 | 948 | 2,210 |
| # of knowledge concepts | 11 | 16 | 8 | 57 | 101 |
| # of interactions | 72,359 | 78,221 | 10,720 | 1,399,470 | 268,709 |
| # of interactions per student | 17.19 | 20.00 | 20.00 | 284.56 | 160.13 |

# E. Implementation Details

For the parameterization of the CSDPM, we employ the extended Graph Transformer architecture from (Dwivedi & Bresson, 2021; Vignac et al., 2023), configuring it with 8 transformer layers, whose hidden dimensions (e.g., MLP, attention heads, and feed-forward layers) are set identically to those in (Vignac et al., 2023). For pretraining the CSDPM, the CSDPM is trained using a uniform transition kernel for diffusion and the AdamW optimizer, with the number of diffusion steps $T$ set as 500, node–edge loss balancing coefficient $\lambda_{ve}$ $(0, 1)$, the batch size $(64, 512)$, dropout rate $(0, 0.5)$, and initial learning rate [1e-5,1e-2] with weight decay tuned via random or grid search strategy. The number of sampled trajectories $\mathcal{D}$ is searched in $\{128, 256, 512\}$. For CSG-KT and CSG-CD, the dimension of the graph pooling for cognitive state representation is searched in $\{8, 16, 32, 64\}$. To configure the training process, we initialize the parameters using Xavier initialization (Glorot & Bengio, 2010) and employ flexible methods such as random, grid, and bayes search& select strategies. For fairness, the hyper-parameter settings of the baseline models have been further tuned using the same tuning strategies to achieve optimal results. All experiments were run on Linux servers equipped with an Intel Xeon Platinum 8352V CPU and NVIDIA RTX 4090 GPUs.

# F. Hyperparameters Analysis

We conducted a sensitivity analysis of some key parameters. We summarize the following observations and conclusions: The optimal node–edge loss balancing coefficient $\lambda_{ve} \in (0, 1)$ was 0.5 for Math1, Math2, and FrcSub, and 0.6 for NIPS34, which has a larger number of nodes yielding a correspondingly greater number of edges. For both CSG-KT and CSG-CD, the optimal graph pooling dimension was 16 for Math1 and Math2, 8 for FrcSub, and 32 for NIPS34.

To further examine the robustness of our reward design, we conducted an ablation study by systematically varying the threshold parameter $\kappa$ of the matching degrees $\mathcal{M}_{\mathcal{V}}$ and $\mathcal{M}_{\mathcal{E}}$, as well as the reward scaling schemes. Specifically, we tested three settings of $\kappa \in \{0.3, 0.5, 0.7\}$, and three reward tuples: (i) a simple linear progression $(1, 2, 3, 4, 5)$, (ii) a steeper linear progression $(1, 3, 5, 7, 9)$, and (iii) an exponential progression $(2, 4, 8, 16, 32)$. Figure 3 summarizes the final AUC and ACC results, where we take the Math2 dataset as a representative example. The combination of $\kappa = 0.5$ with the simple linear reward $(1, 2, 3, 4, 5)$ consistently achieves the best balance between performance and stability. In contrast, exponential scaling tends to amplify the contribution of rare high-level cases, leading to unstable optimization, while the steeper linear scheme introduces uneven signals that bias the model toward intermediate levels. The neutral threshold $\kappa = 0.5$ also proved optimal: a looser setting ($\kappa = 0.3$) misclassifies partially aligned structures, whereas a stricter setting ($\kappa = 0.7$) over-penalizes mid-level structures. In practice, the selected reward tuple yields stable training behavior and consistent performance across datasets, and we apply the same values throughout all experiments without dataset-specific tuning.

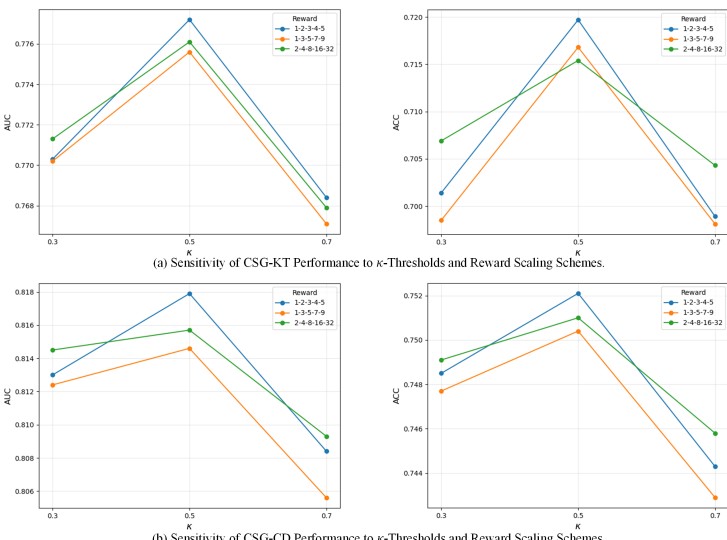

(a) Sensitivity of CSG-KT Performance to $\kappa$-Thresholds and Reward Scaling Schemes.

(b) Sensitivity of CSG-CD Performance to $\kappa$-Thresholds and Reward Scaling Schemes.

*Figure 3.* (**Hyperparameter Study**). Sensitivity of CSG-KT and CSG-CD Performance (AUC and ACC) to $\kappa$-Thresholds and Reward Scaling Schemes.

## G. Inference Time Analysis

The main cost of our CSG is the reverse denoising chain with $T$ steps, each requiring one backbone pass on graph $\mathcal{G}$; thus inference complexity is $O(T \cdot C_{\text{backbone}}(|G|))$. We further provide a more complete comparison by additionally measuring the overall per-graph inference time when performing downstream KT/CD tasks and comparing with representative baselines. Table 7 reports the inference time of CSG for generating a single cognitive-structure graph across datasets with different numbers of nodes. The inference time remains low and scales reasonably with the graph size, demonstrating the practical feasibility of deploying CSG.

*Table 7.* Inference time (ms) for generating a single cognitive-structure graph.

| Dataset | Nodes | Time (ms) |
|---|---|---|
| Math1 | 11 | 2.61 |
| Math2 | 16 | 4.24 |
| FrcSub | 8 | 0.74 |
| NIPS34 | 57 | 25.65 |
| ASSISTment17 | 101 | 44.07 |

We also report the overall per-graph inference time (ms) when CSG is used in downstream KT, and compare it with standard KT baselines. As shown in Table 8, CSG-KT is slower than lightweight classical models such as DKT, but remains within the same order of magnitude as more recent higher-capacity models (e.g., ENAS-KT). Importantly, student cognitive structures do not need to be recomputed in real time; in most educational settings they can be updated periodically (e.g., at the end of a session or unit), which reduces the practical impact of the additional computational cost.

*Table 8.* Per-graph inference time (ms) for KT models.

| Dataset | Nodes | DKT | SAKT | GKT | ENAS-KT | PSI-KT | CSG-KT |
|---|---|---|---|---|---|---|---|
| Math1 | 11 | 0.04 | 0.26 | 6.20 | 1.79 | 3.06 | 2.71 |
| Math2 | 16 | 0.07 | 0.38 | 9.03 | 1.82 | 4.92 | 4.36 |
| FrcSub | 8 | 0.03 | 0.19 | 4.05 | 0.76 | 2.90 | 0.85 |
| NIPS34 | 57 | 0.24 | 1.38 | 32.15 | 15.07 | 31.68 | 26.15 |
| ASSISTment17 | 101 | 0.63 | 2.64 | 69.27 | 28.46 | 66.93 | 63.12 |

Similarly, Table 9 reports the overall per-graph inference time when performing downstream CD with CSG-generated structures, compared with representative CD baselines. The same trend holds: CSG-CD incurs additional cost compared to NCD, while staying in the same order of magnitude as stronger baselines (e.g., RCD).

*Table 9.* Per-graph inference time (ms) for CD models.

| Dataset | Nodes | NCD | RCD | DisenGCD | CSG-CD |
|---|---|---|---|---|---|
| Math1 | 11 | 0.07 | 2.18 | 1.46 | 2.68 |
| Math2 | 16 | 0.08 | 3.18 | 1.67 | 4.29 |
| FrcSub | 8 | 0.06 | 1.59 | 0.34 | 0.81 |
| NIPS34 | 57 | 0.10 | 24.46 | 5.19 | 25.89 |
| ASSISTment17 | 101 | 0.17 | 45.33 | 8.91 | 44.32 |

## H. Simple Example of Cognitive Structure Simulation

Given five questions $q_1$–$q_5$ that assess the concepts *Sine Theorem* and *Cosine Theorem*, we make an idealized assumption: if a question involves only one concept, its weight for that concept is set to 1; if it involves both concepts, the weights for each concept are set to 0.5. Suppose a student $s_i$'s responses to these five questions are recorded as $X_i^5$, as shown in the Table 10 below.

*Table 10.* Example of question weights and student responses.

| Question | Sine Weight | Cosine Weight | Response |
|:---:|:---:|:---:|:---:|
| $q_1$ | 1.0 | 0.0 | Correct |
| $q_2$ | 1.0 | 0.0 | Correct |
| $q_3$ | 0.5 | 0.5 | Correct |
| $q_4$ | 0.5 | 0.5 | Incorrect |
| $q_5$ | 1.0 | 0.0 | Incorrect |

Accordingly, using Eqs.1 and 2, based on interaction records $X_i^5$, we can calculate the student's construction for the concepts *Sine Theorem* and *Cosine Theorem*, the node-level term $f_{UOC}(\textit{Sine Theorem}, X_i^5)$ and the edge-level term $f_{UOR}(\textit{Sine Theorem}, \textit{Cosine Theorem}, X_i^5)$ in the simulated cognitive structure, as follows:

$$f_{UOC}(\textit{Sine Theorem}, X_i^5) = \frac{1.0 \cdot 1.0 + 1.0 \cdot 1.0 + 0.5 \cdot 1.0 + 0.5 \cdot 0 + 1.0 \cdot 0}{1.0 + 1.0 + 0.5 + 0.5 + 1.0} = \frac{2.5}{4.0} = 0.625,$$

$$f_{UOR}(\textit{Sine Theorem}, \textit{Cosine Theorem}, X_i^5) = \frac{0 \cdot (1.0+0) \cdot 0 + 0 \cdot (1.0+0) \cdot 0 + 1.0 \cdot (0.5+0.5) \cdot 1.0 + 1.0 \cdot (0.5+0.5) \cdot 0 + 0 \cdot (1.0+0) \cdot 0}{0 \cdot (1.0+0) + 0 \cdot (1.0+0) + 1.0 \cdot (0.5+0.5) + 1.0 \cdot (0.5+0.5) + 0 \cdot (1.0+0)} = \frac{1.0}{2.0} = 0.5.$$

## I. List of knowledge concepts in Math1

The table below lists the concept names in the Math1 dataset, which are used for the visualization and interpretability analysis.

*Table 11.* List of knowledge concepts in Math1.

| No. | Concept Name |
|:---:|:---|
| 0 | Set |
| 1 | Inequality |
| 2 | Trigonometric function |
| 3 | Logarithm versus exponential |
| 4 | Plane vector |
| 5 | Property of function |
| 6 | Image of function |
| 7 | Spatial imagination |
| 8 | Abstract summarization |
| 9 | Reasoning and demonstration |
| 10 | Calculation |

## J. Limitations and Future Work

CSG leverages diffusion models, which are generally more computationally intensive than classical architectures used in knowledge tracing and cognitive diagnosis, such as LSTMs and GNNs. However, recent advances in accelerating the denoising process of diffusion models (Nichol & Dhariwal, 2021; Liu et al., 2022a; Song et al., 2023; Yin et al., 2024; Rombach et al., 2022; Hang et al., 2025) offer promising avenues to improve efficiency. Moreover, student cognitive structures typically do not require real-time updates, making the added computational cost acceptable in practical settings.

For the simulated cognitive structures, in Stage I, we deliberately use a simple rule-based simulator instead of more complex alternatives such as BKT-/IRT-based simulators or human-elicited cognitive maps. BKT-/IRT-based simulators require training additional models and careful hyperparameter tuning on the same performance data, while expert maps depend on costly manual labeling and are rarely available at scale. By contrast, our weighted-correctness rules provide a transparent, training-free proxy that can be computed directly from existing Q-matrices and logs. We acknowledge that this design may introduce bias, but in our framework these signals are only used for pretraining, and the subsequent SOLO-based RL refinement on real interactions can partially correct such bias. In future work, we plan to explore learned or hybrid simulators that retain interpretability while further reducing reliance on handcrafted rules.

Besides, our current work focuses on a setting standard in KT/CD and many deployed learning systems: a curriculum- or test-defined concept set that is relatively stable within a course or semester (Tyler, 2013; Zhao et al., 2022), and CSG models

how students' cognitive structures over this fixed set evolve across time. The CSG framework is not inherently limited to a flat concept layer. In principle, it can be extended to multi-level or heterogeneous graphs, where nodes represent domains, intermediate concepts, or finer-grained skills, and edges describe relations both within and across levels. New concepts can be incorporated without retraining the entire system. For example, one could **(i)** pretrain the diffusion backbone on a broader ontology and fine-tune it when new concepts appear (Ruiz et al., 2023; Zhuang et al., 2024), or **(ii)** initialize embeddings for new concepts from textual or ontological neighbors (Hamilton et al., 2017) and continue diffusion+RL training with mild regularization to preserve existing structures. More generally, inductive mechanisms such as feature-based initialization, adapter layers, or continual-learning approaches (Zhou et al., 2024a) can be integrated to support dynamically expanding concept sets. We leave a systematic exploration of such multi-level and dynamically evolving extensions as future work.

Besides, our current work focuses on a setting standard in KT/CD and many deployed learning systems: a curriculum- or test-defined concept set that is relatively stable within a course or semester (Tyler, 2013; Zhao et al., 2022), and CSG models how students' cognitive structures over this fixed set evolve across time. The CSG framework, however, is not inherently limited to a flat or fixed concept layer. In principle, it can be extended to multi-level or heterogeneous graphs, where nodes represent domains, intermediate concepts, or finer-grained skills, and edges describe relations both within and across levels. It may also be adapted to broader inductive and cold-start settings involving new students, new questions, or new concepts, which are important for real-world KT/CD deployment. For example, new concepts could be incorporated by pretraining the diffusion backbone on a broader ontology and fine-tuning it when new concepts appear (Ruiz et al., 2023; Zhuang et al., 2024), or by initializing their embeddings from textual or ontological neighbors (Hamilton et al., 2017). Related studies (Liu et al., 2022b; Gao et al., 2023; Liu et al., 2021) on unseen-question prediction and cold-start cognitive diagnosis also provide useful inspiration for extending CSG beyond the current evaluation protocol. More generally, inductive mechanisms such as feature-based initialization, adapter layers, or continual-learning approaches (Zhou et al., 2024a) can be integrated to support dynamically evolving learner, item, and concept spaces. We leave a systematic exploration of these multi-level, inductive, and dynamically evolving extensions as future work.

