# OpenReview forum: "CSG: Cognitive Structure Generation for Intelligent Education"
_ICML.cc/2026/Conference — ICML 2026 regular_

### Official Review · Reviewer_c354 · 2026-03-10

**Soundness:** 3
**Presentation:** 4
**Significance:** 3
**Originality:** 2
**Overall Recommendation:** 4
**Confidence:** 4

**Summary:**

This paper proposes CSG (Cognitive Structure Generation), a framework for explicitly modeling students’ cognitive structures from interaction data. The method formulates student modeling as a structure generation problem rather than a direct prediction task. Specifically, the approach first constructs simulated cognitive structures based on students’ historical interactions with exercises and concept annotations. A Cognitive Structure Diffusion Probabilistic Model (CSDPM) is then introduced to generate cognitive structure graphs consisting of concept-level node states and relation-level edge states. To further refine the generated structures, the paper incorporates a reinforcement learning optimization stage guided by SOLO taxonomy–based hierarchical rewards. The generated cognitive structures are subsequently used as representations for downstream tasks such as knowledge tracing and cognitive diagnosis. Extensive experiments on multiple benchmark datasets are conducted to evaluate the effectiveness of the proposed framework compared with existing student modeling methods.

**Compliance With Llm Reviewing Policy:**

Affirmed.

**Key Questions For Authors:**

1. **What is the semantic meaning of the edges in the generated cognitive structure graph?** The paper models relations between concepts as edges in a graph, but the exact semantic interpretation of these relations is not clearly defined. While the model approximates relation states using heuristic rules based on co-occurrence and performance, it remains unclear whether the edges represent prerequisite relationships, conceptual dependencies, or merely statistical correlations derived from student interaction data. Clarifying the semantic meaning of these relations and explaining how the model ensures educational interpretability would help strengthen the interpretability claim of the framework.
2. **How does the SOLO taxonomy–based reward function explicitly map to graph structural properties?** The reinforcement learning stage introduces a reward function based on SOLO taxonomy levels to guide the generation of cognitively meaningful structures. However, the paper does not clearly explain the explicit mathematical mapping between SOLO levels and specific structural characteristics (e.g., node centrality, edge density, or subgraph patterns) of the generated graphs. A more explicit explanation of this mapping and additional analysis of how the reward specifically influences these structural properties would help clarify the effectiveness of the reinforcement learning stage.
3. **While the appendix provides generation rules, how can it be proven that such rule-based simulation truly represents the complex evolution of human cognitive structures?** In the first training stage, the diffusion model is trained on simulated cognitive structures constructed using heuristic rules. However, although the appendix provides these generation rules, the paper does not provide sufficient evidence or validation to prove that such rule-based simulation can truly represent the complex and latent cognitive evolution of real human students. Providing a clearer empirical validation of the simulated data distribution and discussing how potential gaps between rules and real cognitive processes might affect the diffusion model's reliability would help justify the pretraining process.

**Limitations:**

yes

**Strengths And Weaknesses:**

**Paper Strengths**

1. The paper studies an important problem in intelligent education, namely explicitly modeling students’ cognitive structures rather than only predicting performance outcomes. By reformulating student modeling as a cognitive structure generation task, the work attempts to bridge the gap between traditional predictive approaches (e.g., knowledge tracing and cognitive diagnosis) and educational theories that emphasize structural knowledge representation.
2. The proposed framework integrates graph-based modeling with diffusion generative techniques to produce student-specific cognitive structures. Compared with conventional representation-learning approaches that output latent vectors, the graph generation formulation potentially provides more interpretable structural representations of concept mastery and inter-concept relations.
3. The method introduces a two-stage training strategy consisting of diffusion-based pretraining followed by reinforcement learning optimization using SOLO taxonomy rewards. This design attempts to incorporate educational cognitive hierarchy into the learning objective, which is conceptually interesting and reflects an effort to align machine learning models with educational assessment theory.

**Weaknesses**

1. In terms of methodological novelty, although the paper introduces a diffusion-based framework for cognitive structure generation, many components are adapted from existing techniques such as graph diffusion models, graph transformers, and reinforcement learning with policy gradients. The proposed framework appears largely as a combination of these components rather than a fundamentally new modeling paradigm specifically tailored for student modeling.
2. The definition of the cognitive structure graph is not sufficiently rigorous. While nodes represent concept-level mastery states and edges represent relations between concepts, the semantic meaning of these relations is not clearly specified (e.g., prerequisite dependency, co-occurrence relation, or latent correlation). Without a clear definition, it is difficult to determine whether the generated structures truly correspond to meaningful cognitive structures in educational theory.
3. The pretraining stage relies on simulated cognitive structures constructed from heuristic rules derived from interaction histories. However, the paper does not provide a detailed justification or validation of the simulation procedure. If the simulated structures deviate from real cognitive structures, the diffusion model may learn biases introduced by the heuristic generation process.

---

> ### Author Rebuttal · Authors · 2026-03-31
>
> > Weakness 1: Novelty
>
> Thank you. We agree our novelty is not a new generic diffusion model, Graph Transformer, or RL algorithm in isolation. Rather, it lies in a new student-modeling formulation and its task-specific integration. Prior methods usually couple representation learning with downstream prediction, where graphs mainly serve as auxiliary bias inside KT/CD predictors. In contrast, CSG formulates a new task: generating learner-specific, explicit cognitive-structure graphs from interaction histories, decoupled from any single predictor and reusable across KT and CD. Here, graph diffusion enables explicit graph generation, the backbone conditions on interaction histories, and SOLO-guided policy optimization refines the generator when ground-truth structures are unavailable. Thus, the contribution is a new formulation plus an integrated framework tailored to it, enabling explicit structure generation, predictor decoupling, cross-task reuse, and better interpretability.
>
> > Weakness 2 and Q1: Edge semantics
>
> Thank you for the comment. In this work, we adopt a general definition of cognitive structure, reflecting the learner’s internal organization of concepts and their relations [1]. The edges represent data-driven latent inter-concept associations inferred from student interaction data, rather than a specific predefined relation type (e.g., prerequisite dependencies). Since explicit relation annotations are typically unavailable, we use concept co-occurrence as an implicit proxy and combine interaction history to estimate whether such associations are supported by observed learning behaviors. In this sense, the edges capture interaction-supported conceptual associations that can correspond to different semantic relations when labeled data are available. Adapting CSG to specific relation semantics involves adjusting the relation construction (Stage I, Eq. 1–2) according to the chosen definition or proxy; otherwise, it aligns with labeled relations without modification.
>
> > Weakness 3 and Q3: Rule-based simulation bias
>
> Thank you. We agree Stage-I simulation is not a full or exact representation of real human cognitive evolution. It is only a theory-guided proxy providing an initial supervisory signal when real cognitive structures are unobservable. Importantly, CSG is designed to mitigate rather than preserve this bias: Stage II does not reconstruct the simulated structures, but optimizes a SOLO-based reward from alignment between the generated structure and the next observed interaction, pushing the generator beyond handcrafted assumptions toward structures better supported by learning evidence. This is consistent with the ablations: full CSG consistently outperforms pretraining-only or weaker variants, and the gain over V5 shows the value of SOLO-based refinement beyond Stage-I initialization. To address this more directly, we add a human-evaluation experiment by extending Table 5 with Stage-I simulated CS:
>
> |Dataset|Proxy&Generator|Jaccard↑|Graph Edit Distance↓|
> |---|---|---:|---:|
> |FrcSub|Random|0.18|0.87|
> || GPT-5|0.65|0.30|
> ||Llama-3-70B|0.39|0.61|
> ||Simulated CS|0.46|0.39|
> ||CSG|**0.79**|**0.15**|
> |Math2|Random|0.14| 0.89|
> ||GPT-5|0.54|0.43|
> ||Llama-3-70B|0.34|0.67|
> ||Simulated CS|0.45|0.54|
> ||CSG|**0.69**|**0.21**|
>
> The results show that Stage-I simulated CS already has non-trivial agreement with human judgment, indicating an informative rather than arbitrary initialization; more importantly, Stage II substantially improves it and outperforms LLMs. We will add these results and discussion in the revision. We will also discuss richer priors, e.g., BKT-/IRT-based simulators or human-elicited structures, as future extensions, though they introduce extra fitting cost, tuning, and pipeline complexity.
>
> > Q2: SOLO reward mapping
>
> Thank you. Our intention is not to map SOLO levels to specific graph statistics such as node centrality, edge density, or predefined motifs. Instead, we use SOLO as an educationally grounded scaffold to define a hierarchical reward for cognitive-structure generation. Cognitive structure theory [1] views learning as gradually constructing an internal organization of concepts and their relations; SOLO provides a simplified abstraction of this progression. Thus, the reward serves as a high-level optimization signal guiding the generation process toward graphs whose overall organization better reflects a progression from fragmented to more integrated structures, rather than enforcing any single structural metric. As in many RL-based generative settings, a scalar reward mainly functions as a global evaluative objective rather than directly supervising individual graph properties [2,3].
>
> ---
>
> [1] Ausubel D P, Novak J D, Hanesian H. Educational psychology: A cognitive view[J]. 1978.
>
> [2] Ouyang L, et al. Training language models to follow instructions with human feedback. NeurIPS 2022.
>
> [3] Liu Y, et al. Graph diffusion policy optimization. NeurIPS 2024.

---

> > ### Author Rebuttal · Reviewer_c354 · 2026-04-03
> >
> > After reading the authors' response, I decided to maintain my score.

---

### Official Review · Reviewer_KXxb · 2026-03-13

**Soundness:** 3
**Presentation:** 4
**Significance:** 3
**Originality:** 2
**Overall Recommendation:** 5
**Confidence:** 3

**Summary:**

The authors present a diffusion-based architecture to generate personalized knowledge component (KC) graphs. They show on several datasets that they outperform either knowledge tracing or cognitive diagnosis techniques. The approach is original, the paper is well written.

**Compliance With Llm Reviewing Policy:**

Affirmed.

**Final Justification:**

I carefully read the paper, and DiffCog and DiffuQKT, the baselines that were initially missing in the paper. The approach of this paper is unique as they are not only generating the latent state of student knowledge using diffusion; but a personalized knowledge component graph. It resonates with my own ongoing work (which BTW is not cited in the paper). This paper is solid, and would benefit from more abundant discussion to compare with its competitors, which we started to see in the rebuttal, which addressed my concerns. This is why I would like to recommend acceptance.

**Key Questions For Authors:**

- Why didn't you at least compare the predictive performance of the proposed approach with DiffCog? In your current baselines some approaches also rely on an implicit latent representation (like SAKT).
- Why didn't you include Assistments 2009 in the analysis, while it is a more famous dataset?

**Limitations:**

Yes

**Strengths And Weaknesses:**

I previously reviewed this paper. This is a (I thought) unique approach that draws a link between knowledge tracing and cognitive diagnosis, two popular research communities in the literature. It also addresses the shortcomings that not everyone may have the same representations for a domain.

The mathematical description of the proposed approach is very clear. However, the novelty compared to the competitor DiffCog is unclear. The authors state that their approach focuses on an explicit personalized KC graph compared to the latent graph of DiffCog. Still, it is unclear why the authors do not include DiffCog in their benchmark, as both papers seem to share a dataset (NeurIPS 2020) with DiffCog having a better performance. To put it clearly, the authors acknowledge that DiffCog exists, but their arguments not to include it in the benchmark are insufficient. It is still useful to know the predictive power of a less interpretable model, as it is the "closest neighbor" method that can perform well on the same task.

To me, the link with SOLO taxonomy is a bit far-fetched: the fact that there would be exactly 5 levels is arbitrary. I understand that the authors chose it because this taxonomy was chosen by PISA.

---

> ### Author Rebuttal · Authors · 2026-03-31
>
> Thank you very much for reviewing this work again. We sincerely appreciate your continued engagement with the paper. Your previous feedback helped us improve the framing and clarity of the manuscript.
>
> > Weakness 1 and Question 1: Compared to the diffusion-based methods
>
> Thank you for this important suggestion. We agree that diffusion-based methods are the most relevant neighboring baselines. Following the reviewer’s suggestion, we have now added direct comparisons with DiffCog [1] for CD and additional DiffuQKT [2] for KT on the shared benchmark datasets. We also clarify the NIPS setting for fairness: DiffCog reports results on the full NIPS2020 dataset [3], whereas our experiments follow the widely used Task34 subset of NIPS2020 adopted in prior studies [4]. Thus, the NIPS results are related but not based on exactly the same benchmark split.
>
> |Task|Method||Math1|||Math2|||FrcSub|||NIPS|||Assist17||
> |---|---|---:|---:|---:|---:|---:|---:|---:|---:|---:|---:|---:|---:|---:|---:|---:|
> || |AUC|ACC|RMSE|AUC|ACC|RMSE|AUC|ACC|RMSE|AUC|ACC|RMSE|AUC|ACC|RMSE|
> |KT|DiffuQKT|0.8116|0.7347|0.4333|0.7716|0.7098|0.4406|0.8519|0.7889|0.3225|0.7300|0.6695|0.4510|0.7756|0.7579|0.4394|
> ||CSG-KT|0.8220|0.7412|0.4283|0.7772|0.7197|0.4390|0.8636|0.8022|0.3192|0.7413|0.6757|0.4511|0.7963|0.7792|0.4313|
> |CD|DiffCog|0.7892|0.7115|0.4403|0.7766|0.7038|0.4421|0.8623|0.7873|0.3928|0.7755|0.7098|0.4401|0.7772|0.7083|0.4419|
> ||CSG-CD|0.8133|0.7710|0.3987|0.8179|0.7521|0.4270|0.8699|0.8451|0.3152|0.8036|0.7507|0.4242|0.8003|0.7386|0.4320|
>
> Overall, CSG achieves competitive predictive performance while maintaining an explicit and interpretable cognitive structure compared to latent-state diffusion methods.
>
> > Weakness 2: To me, the link with SOLO taxonomy is a bit far-fetched: the fact that there would be exactly 5 levels is arbitrary. I understand that the authors chose it because this taxonomy was chosen by PISA.
>
> Thank you for the insightful comment. We agree that, from a purely modeling perspective, the number of reward levels need not be exactly five. Our use of five levels is not meant to claim that student cognition intrinsically evolves in exactly five discrete stages. Rather, it is a theory-informed design choice: since the SOLO taxonomy [5] itself defines five levels, we use it as an educationally meaningful scaffold for hierarchical reward design. Actually, both our CSG framework and downstream student modeling tasks such as KT and CD are ultimately intended to serve education, so their outputs should also remain interpretable at the educational level.
>
> More broadly, we view this as an initial attempt to incorporate educational theory into cognitive-structure generation, rather than as a claim that SOLO is the only or uniquely correct discretization. Future work will explore more adaptive reward formulations that better capture developmentally meaningful learning patterns and their variability across students.
>
> > Question 2: Why didn't you include Assistments 2009 in the analysis, while it is a more famous dataset?
>
> Thank you for the suggestion. ASSISTments2009 is indeed a classic and influential benchmark in KT.  We would like to clarify the reason that we chose the ASSISTments17 as one of our datasets.
>
> First, ASSISTments2009 and ASSISTments17 come from the same platform and have a comparable concept scale, making ASSISTments17 a representative large-scale benchmark for our study.
>
> Second, from the perspective of realistic educational settings, questions assessing multiple concepts are more common and also more aligned with the motivation of our framework. We therefore prioritized datasets such as Math and ASSISTment17, where one-to-many question–concept associations are more prevalent. By contrast, ASSISTments2009 contains relatively fewer such cases and is dominated more by one-to-one mappings, which leads to limited structural diversity, making it less informative for evaluating multi-concept structural modeling.
>
> ---
>
> [1] Zhao G, et al. A diffusion-based cognitive diagnosis framework for robust learner assessment.TLT 2024.
>
> [2] Yu F, et al. DiffuQKT: A Diffusion-Based Approach for Improved Question Representation in Knowledge Tracing.MM 2025.
>
> [3] Wang Z, et al. Instructions and guide for diagnostic questions: The neurips 2020 education challenge. arXiv:2007.12061, 2020.
>
> [4] Liu Z, et al. pyKT: a python library to benchmark deep learning based knowledge tracing models. NeurIPS 2022.
>
> [5] Biggs J B, Collis K F. Evaluating the quality of learning: The SOLO taxonomy (Structure of the Observed Learning Outcome)[M]. Academic press, 2014.

---

> > ### Author Rebuttal · Reviewer_KXxb · 2026-04-04
> >
> > Thanks for providing these extra experiments. I would like to upgrade my score, but I still have a couple follow-up questions. Please note that [ASSISTments 2017](https://sites.google.com/view/assistmentsdatamining/data-mining-competition-2017) challenge is originally *very different* from knowledge tracing as this is predicting a long-term outcome.
> > > Long term outcome being modeled: The task in this competition is to develop a cross-validated prediction model that is able to use middle-school data to predict whether the students (who have now finished college) pursue a career in STEM fields (1) or not (0).
> >
> > Still, it may be possible that the middle-school data can be used for a knowledge tracing task; but I think this explains why 2017 is not seen in traditional benchmarks such as pyKT, although it exists in some KT papers.
> >
> > **Q1.** Could the authors please confirm that they are aware of this fact and that they used the right column for prediction in ASSISTments 2017?
> >
> > Thanks for clarifying the differences between NIPS 2020 data. It would be nice to have the right splits for the paper. I understood the multiple-skill argument for ASSISTments 2009 but as there are several other KT datasets, I would still encourage the authors to consider either ASSISTments 2009/2012/Eedi/EdNet which are in the DiffuQKT paper (the latter two being massive) or other datasets encountered in pyKT.
> >
> > **Q2.** I see now that the original DiffCog paper does not share any code. Did the authors reimplement it from scratch for the comparison, just for the rebuttal? Did the authors help themselves with generative AI for that?

---

> > > ### Author Response · Authors · 2026-04-04
> > >
> > > Dear Reviewer KXxb,
> > >
> > > Thank you very much for reading our rebuttal and the constructive suggestions. We are more than happy to clarify the follow-up questions.
> > >
> > > > Q1. ASSISTments2017
> > >
> > > We thank the reviewer for attention to details for the ASSISTments 2017 dataset. We are aware that the dataset was originally used for modeling long-term outcomes on whether a student will pursue a STEM career. Because it contains per-interaction response labels, other researchers and us have repurposed the dataset for KT / CD tasks. Specifically, we use the per-interaction `correct` field as the response label and do not use the `isSTEM` field for all experiments. We will make this explicit in the paper to avoid any potential confusion.
> > >
> > > We also appreciate the reviewer’s suggestion on datasets. We will in future further expand the benchmark coverage by considering more KT datasets used in recent works such as Ednet and KDD Cup.
> > >
> > > > Q2. DiffCog Implementation
> > >
> > > Thank you for raising this question and there is already an open-source implementation of DiffCog available on GitHub from the authors: https://github.com/ghzha0/DiffCog
> > >
> > > The DiffCog paper indeed only mentions “Our code will be published after this article is accepted” and we located the above repository via Google search. We used this publicly available implementation for our rebuttal experiments. We did not reimplement DiffCog from scratch, nor did we use generative AI in this process. We will clarify this explicitly in the revised version of the paper for transparency.
> > >
> > > Thank you again for raising your recommendation and the helpful feedback for our work.
> > >
> > > Regards,
> > >
> > > Authors

---

### Official Review · Reviewer_2Ahs · 2026-03-14

**Soundness:** 2
**Presentation:** 3
**Significance:** 3
**Originality:** 3
**Overall Recommendation:** 4
**Confidence:** 4

**Summary:**

This paper focuses on the explicit modeling and generation of students' cognitive structures (CS) in intelligent education. Traditional methods (such as KT and CD models) often only focus on the degree of concept mastery, while neglecting or weakening the modeling of relationships between concepts. This paper proposes a cognitive structure generation method based on a diffusion probability model (CSDPM), which learns the distribution of students' cognitive structures through a graph diffusion model. Simultaneously, the authors designed a regularized cognitive structure simulation mechanism based on response logs and Q matrices for pre-training, and optimized the generation process by combining reinforcement learning and a hierarchical reward function based on SOLO taxonomy. Experimental results show that the generated cognitive structures can serve as effective student representations and improve the performance of knowledge tracking and cognitive diagnosis tasks.

**Compliance With Llm Reviewing Policy:**

Affirmed.

**Key Questions For Authors:**

Q1: Some student modeling methods have also used graph structures to represent knowledge concepts and their relationships (such as RCD). Could the author further clarify the essential differences between the cognitive structure generated in this paper and the concept graph structures in these methods in terms of modeling goals and expressive capabilities?

Q2: Diffusion models typically involve T-step iterative denoising, and this paper also involves the generation of graph structures. As the number of knowledge points increases, the complexity of edge set generation increases. How scalable is this model when dealing with a larger number of concept sets?

**Limitations:**

The paper heavily relies on the simulated cognitive structure generated based on rules in the first stage. The authors should discuss further: if the simulated logic used in pre-training deviates fundamentally from real human cognitive processes, will the generative model further amplify these biases through reinforcement learning? It is recommended to add a discussion on the "robustness of the model to simulated rules."

**Strengths And Weaknesses:**

Strengths

1.	This paper takes a novel research perspective on cognitive structure generation, with a relatively clear overall structure and a relatively systematic description of the methodology.

2.	The paper proposes a two-stage learning framework that includes cognitive structure simulation, diffusion generative model training, and reinforcement learning optimization, with clear logical connections between the modules.

3.	Experiments were conducted on multiple publicly available educational datasets, and the generated cognitive structures were applied to knowledge tracing and cognitive diagnostic tasks. The effectiveness of the method was verified by comparing it with various baseline models.

Weaknesses

1.	The simulated construction of cognitive structures lacks sufficient theoretical basis, especially since the first stage of pre-training heavily relies on rule-based simulation data. Although experiments have proven its effectiveness, if the simulation logic deviates significantly from actual learning patterns, it may introduce pre-existing biases into the model. This paper does not delve deeply enough into this risk.

2.	Although this paper proposes generating cognitive structures, the differences and advantages between it and existing graph structure learning modeling methods still need to be further clarified at the theoretical level.

3.	The experiments in this paper still demonstrate the effectiveness of the proposed method through performance comparison. Whether the generated cognitive structures can support educational applications such as teaching diagnosis has not yet been fully analyzed through experiments or case studies.

---

> ### Author Rebuttal · Authors · 2026-03-31
>
> > Weakness 1 and Limitation: Robustness of rule-based simulation
>
> Thank you for this important concern. To address it directly, we add a human-evaluation experiment by extending Table 5 with the Stage-I simulated cognitive structures. We compare simulated CS against human-annotated structures using Jaccard and GED on FrcSub and Math2:
>
> |Dataset|Proxy&Generator|Jaccard↑|Graph Edit Distance↓|
> |---|---|---:|---:|
> |FrcSub|Random|0.18|0.87|
> ||GPT-5|0.65|0.30|
> ||Llama-3-70B|0.39|0.61|
> ||Simulated CS|0.46|0.39|
> ||CSG|**0.79**|**0.15**|
> |Math2|Random|0.14|0.89|
> ||GPT-5|0.54|0.43|
> ||Llama-3-70B|0.34|0.67|
> ||Simulated CS|0.45|0.54|
> ||CSG|**0.69**|**0.21**|
>
> These results show that Stage-I simulated CS already has non-trivial agreement with human judgment, so the rules provide an informative rather than arbitrary initialization. More importantly, Stage II substantially improves it and even outperforms LLMs. We will add these results and discussion in the revision. More broadly, we will discuss richer priors, e.g., BKT-/IRT-based simulators or human-elicited structures, as future extensions to further reduce dependence on handcrafted rules and potential Stage-I bias, though they also introduce extra fitting cost, tuning, and pipeline complexity.
>
> > Weakness 2 and Question 1: Difference from prior graph-based methods
>
> Thank you for this important comment. Existing graph-based student modeling methods usually use predefined concept graphs/maps [1] or heterogeneous interaction graphs [2,3] to improve estimation of latent mastery states. Their graphs mainly serve task-specific KT/CD prediction, and the learned student representations are still largely latent. In contrast, CSG explicitly generates a learner-specific cognitive structure from interaction history, where nodes denote concept-construction states and edges denote inter-concept relation-construction states. These structures are dynamic, individualized, decoupled from the predictor, and reusable across KT/CD tasks. Hence, CSG targets explicit, interpretable structure generation rather than only better latent representation learning; Appendix G provides corresponding cases.
>
> > Weakness 3: Educational diagnostic value beyond predictive gains
>
>
> Thank you for this valuable comment. We mainly validate CSG through downstream prediction because cognitive structure is latent and direct ground truth is unavailable. Still, Appendix G provides initial evidence for diagnostic value: (i) visualization and temporal-evolution cases show how concept states and relation states explain different response outcomes and structural development over time; (ii) the human study shows that CSG-generated structures agree more with expert annotations than LLM-generated ones. These results suggest the graphs are not only predictive but also provide actionable signals for teaching diagnosis, such as identifying missing prerequisite relations or unstable concept mastery. We agree this is not yet full validation in real educational applications (e.g., teaching diagnosis), and will clarify this scope and position application-level validation as important future work.
>
> > Question 2: Scalability to larger concept sets
>
> Thank you for this important question. We agree scalability becomes more important as the number of concepts grows, since diffusion requires iterative denoising and graph generation becomes more expensive with larger node/edge spaces. In the current paper, CSG is evaluated on datasets with different graph sizes and consistently outperforms strong KT/CD baselines from smaller to larger settings; Appendix H also reports inference time per generated graph. Recent diffusion advances can further accelerate our framework: latent diffusion [4] reduces cost by operating in compressed spaces, and improved noise schedules [5] improve training efficiency. These techniques are complementary and can be integrated into future CSG versions. Moreover, once pretrained, CSG amortizes graph-structure learning across downstream KT/CD tasks because the generated structures can be reused. We agree that scaling to much larger concept sets remains important for future work, and hierarchical/group-wise decomposition is a promising direction.
>
> ---
>
> [1] Nakagawa H, et al. Graph-based knowledge tracing: modeling student proficiency using graph neural network.WWW 2019.
>
> [2] Gao W, et al. RCD: Relation map driven cognitive diagnosis for intelligent education systems.SIGIR 2021.
>
> [3] Yang S, et al. DisenGCD: A meta multigraph-assisted disentangled graph learning framework for cognitive diagnosis.NeurIPS 2024.
>
> [4] Rombach R, et al. High-resolution image synthesis with latent diffusion models. CVPR 2022.
>
> [5] Hang T, et al. Improved noise schedule for diffusion training. CVPR 2025.

---

> > ### Author Rebuttal · Reviewer_2Ahs · 2026-04-04
> >
> > The author has basically solved my questions.

---

### Official Review · Reviewer_L91o · 2026-03-25

**Soundness:** 3
**Presentation:** 3
**Significance:** 3
**Originality:** 3
**Overall Recommendation:** 4
**Confidence:** 2

**Summary:**

This paper proposes Cognitive Structure Generation (CSG), a task-agnostic framework that explicitly models CS through generative modeling. CSG first pretrains a Cognitive Structure Diffusion Probabilistic Model (CSDPM) and then applies reinforcement learning with SOLO-based hierarchical rewards to capture plausible patterns of cognitive development.

**Compliance With Llm Reviewing Policy:**

Affirmed.

**Final Justification:**

I thank the authors for their response and have increased my overall score from 3 to 4.

**Key Questions For Authors:**

1.	Can an ablation be provided for each of the 5 SOLO-based rewards?
2.	Representative CSG graph should be included in the main text for interpretability analysis

**Limitations:**

Provided in appendix.

**Strengths And Weaknesses:**

-> Strengths:

1.	Proposes a task-agnostic framework (CSG) to explicitly model CS through generative modeling, leading to an interpretable analysis of a student’s construction of concepts and inter-concept relations.
2.	CSG first pretrains a Cognitive Structure Diffusion Probabilistic Model (CSDPM) and then applies reinforcement learning with SOLO-based hierarchical rewards to capture plausible patterns of cognitive development.
3.	CS representations are decoupled from the downstream task (KT/CD) to enable transferable CS representations.
4.	Experiments on five real-world datasets show that CSG yields more comprehensive representations and better performance than all baselines.

-> Weaknesses:

1.	Trends in tables 1 and 2 are perfect. The proposed method outperforms all baselines/ablations across all 5 datasets without exception. Are the baselines competitive/SOTA? Is the comparison fair? CSG is parameterized by a graph transformer model (line 230), whereas other baselines might use simpler models. Are the performance gains from better methodology or an increase in parameters of the underlying model improving expressiveness? Authors should clarify how a fair comparison is ensured.
2.	Questions in KT datasets might not have associated ground-truth KCs/concepts. Do the rule-based functions to simulate CS require questions to have associated concepts? How is this method applied if no associated concepts are available?
3.	Authors should provide a clearer distinction between which work/steps (DPM, rule-based functions to simulate CS, etc.) are novel and newly introduced in this paper, and which are existing. This distinction is not clear in Sections 3.2 and 3.3.
4.	How reliable and accurate (wrt human evaluation) are the simulated CS obtained via the rule-based functions?
5.	How scalable is this method wrt the size of the graph? What’s the training and inference time complexity? What happens if the graph evolves (new concepts, etc)?
6.	Training split is strictly disjoint (line 376). Disjoint over student-question interactions, over questions, or over students? Is generalization tested with new questions and/or new students in the test set?

---

> ### Author Rebuttal · Authors · 2026-03-31
>
> > Weakness 1: Fair comparison
>
> Thank you. The Graph Transformer is used only in CSDPM, not in the downstream KT/CD predictor; after generation, the learned structures are fed to simple MLPs (Eq. 14–15). Thus, gains are more likely from better cognitive-structure modeling than from a stronger predictor. We compare with strong baselines, e.g., DiffuQKT [1], DisenGCD [2], PSI-KT [3], and DiffCog [4]. All methods use the same protocol, splits, metrics, tuning, and convergence criterion.
>
> > Weakness 2: Dependence on annotations
>
> Thank you. Stage I leverages question–concept associations, i.e., concept annotations or an equivalent Q-matrix. This is standard in many CD/KT methods [1,5,6,7]. All five datasets here provide such annotations. When unavailable, they could first be mined or learned from data [8,9]. We adopt the annotated setting to focus on cognitive-structure generation and avoid extra uncertainty from concept discovery.
>
> > Weakness 3: Novelty versus existing components
>
> Thank you. Our novelty is not the diffusion model, GNN backbone, or generic RL algorithm alone. Our contribution is to introduce cognitive structure generation (CSG) as a new task, and to instantiate it as a two-stage framework: (i) formulating student cognitive structure as learner-specific graph generation rather than latent-state prediction; (ii) designing a rule-based simulation proxy for Stage-I pretraining when real structures are unavailable; and (iii) introducing a SOLO-based hierarchical reward in Stage II to guide the generator toward developmentally meaningful structures. To our knowledge, this unified CSG formulation has not appeared in prior student modeling work.
>
> > Weakness 4: Reliability of simulated CS
>
> Thank you. The simulated cognitive structures are not treated as ground truth or as fully accurate w.r.t. human judgment. They serve only as a theory-guided, interpretable proxy for Stage I; Stage II refines the generator beyond heuristic bias. Human evaluation targets the final CSG-generated structures, not the Stage-I proxies. Appendix G shows that optimized CSG outputs agree better with expert annotations than LLM-generated structures, where we add new reliability study, please see our responses to Reviewers 2Ahs and c354.
>
> > Weakness 5: Scalability and evolving graphs
>
> Thank you. The main cost is the reverse denoising chain with T steps, each requiring one backbone pass on graph G; thus inference complexity is $O(T·C_{backbone}(|G|))$. Appendix H reports practical inference times on different graph sizes, suggesting reasonable scalability for KT/CD settings. We will state the complexity more explicitly in the revision. The current paper assumes a fixed concept universe and Q-matrix; evolving graphs (e.g., new concepts) are future work (Appendix K). The graph scale matches typical educational measurement settings [10,11].
>
> > Weakness 6: Meaning of split
>
> Thank you. The split is interaction-level disjoint: student–question interaction records are partitioned into train/validation/test sets, and no response record appears in more than one split. Thus, the same student or question may appear across splits, but not the same interaction instance.
>
> > Question 1: Ablation for the five SOLO rewards
>
> Thank you. $r_1,…,r_5$ are not five independent reward terms, but five ordered outcomes of one piecewise SOLO-based reward function $r_{solo}(G_0)$. Thus, separate ablations would not match the design. Instead, we test SOLO through: (i) Table 3 variants without SOLO-based optimization (e.g., generic reward or no Stage-II reward optimization), and (ii) Appendix F sensitivity analyses on threshold κ and reward scaling tuples.
>
> > Question 2
>
> Thank you. We agree the representative CSG graphs were placed in the appendix due to space limits. In revision, we will move them into the main text.
>
> ---
>
> [1] Yu F,et al.DiffuQKT: A Diffusion-Based Approach for Improved Question Representation in Knowledge Tracing.MM2025.
>
> [2] Yang S,et al.DisenGCD: A meta multigraph-assisted disentangled graph learning framework for cognitive diagnosis.NeurIPS2024.
>
> [3] Zhou H,et al.Predictive, scalable and interpretable knowledge tracing on structured domains.ICLR2024.
>
> [4] Zhao G,et al. A diffusion-based cognitive diagnosis framework for robust learner assessment. TLT2024.
>
> [5] Wang F,et al.Neural cognitive diagnosis for intelligent education systems.AAAI2020.
>
> [6] Shen J,et al.Capturing homogeneous influence among students: Hypergraph cognitive diagnosis for intelligent education systems.KDD2024.
>
> [7] Yin Y, et al.Tracing knowledge instead of patterns: Stable knowledge tracing with diagnostic transformer.WWW2023.
>
> [8] Yang H, et al.A novel quantitative relationship neural network for explainable cognitive diagnosis model.KBS2022.
>
> [9] Chen X, et al.Disentangling cognitive diagnosis with limited exercise labels.NeurIPS2023.
>
> [10] Cognitive diagnostic assessment for education:Theory and applications.2007.
>
> [11] Doignon J P,et al.Knowledge spaces.2012.

---

> > ### Author Rebuttal · Reviewer_L91o · 2026-04-04
> >
> > I thank the authors for their response and have increased my overall score from 3 to 4.
> >
> > The split at the student-question interaction level limits the generalizability of the results to new students and new questions. Existing KT papers (e.g., Open-ended Knowledge Tracing: OKT) report generalizability results to new students (split on students) and to new questions.

---

> > > ### Author Response · Authors · 2026-04-05
> > >
> > > Dear Reviewer L91o,
> > >
> > > Thank you very much for the careful follow-up. We are especially grateful for your recognition of the paper and for increasing your overall score.
> > >
> > > We fully agree with your point that generalization and cold-start issues involving new students, new questions, and even new concepts are highly meaningful in learner modeling tasks such as KT and CD, and are also critical for deployment. We also recognize that this remains an important limitation in KT evaluation more broadly, and your suggestion is very valuable.
> > >
> > > We will take this direction seriously in our future work. Relevant prior studies have already explored broader generalization settings, such as prediction on unseen questions (e.g., OKT [1], EKT [2]), and cold-start cognitive diagnosis (TechCD [3]). These works provide valuable inspiration for extending CSG beyond the current protocol, and we sincerely thank you for highlighting this important direction.
> > >
> > > Thank you again for raising your recommendation and the helpful feedback for our work.
> > >
> > > Regards,
> > >
> > > Authors
> > >
> > > —
> > >
> > > References
> > >
> > > [1] Liu N, Wang Z, Baraniuk R, et al. Open-ended knowledge tracing for computer science education. ACL 2022: 3849-3862.
> > >
> > > [2] Liu Q, Huang Z, Yin Y, et al. Ekt: Exercise-aware knowledge tracing for student performance prediction.TKDE  2019, 33(1): 100-115.
> > >
> > > [3] Gao W, Wang H, Liu Q, et al. Leveraging transferable knowledge concept graph embedding for cold-start cognitive diagnosis. SIGIR 2023: 983-992.

---

### Decision · Program_Chairs · 2026-04-30

**Decision:**

Accept (regular)

**Comment:**

This paper proposes Cognitive Structure Generation (CSG), a task-agnostic framework that explicitly models cognitive structure through generative modeling. Experiments on both knowledge tracing (KT) and cognitive diagnosis (CD) demonstrate its effectiveness. During the rebuttal, the authors clarified concerns regarding its differences from graph-based and diffusion-based methods, as well as its scalability to larger concept sets. Please revise the paper carefully and incorporate the human evaluation results into the final version.